# ComFe: An Interpretable Head for Vision Transformers

**Evelyn J. Mannix**                                    *evelyn.mannix@unimelb.edu.au*
*School of Mathematics and Statistics*
*University of Melbourne*

**Liam Hodgkinson**                                    *lhodgkinson@unimelb.edu.au*
*School of Mathematics and Statistics*
*University of Melbourne*

**Howard Bondell**                                    *howard.bondell@unimelb.edu.au*
*School of Mathematics and Statistics*
*University of Melbourne*

**Reviewed on OpenReview:** *https://openreview.net/forum?id=cI4wrDYFqE*

## Abstract

Interpretable computer vision models explain their classifications through comparing the distances between the local embeddings of an image and a set of prototypes that represent the training data. However, these approaches introduce additional hyper-parameters that need to be tuned to apply to new datasets, scale poorly, and are more computationally intensive to train in comparison to *black-box* approaches. In this work, we introduce **Com**ponent **Fe**atures (ComFe), a highly scalable interpretable-by-design image classification head for pretrained Vision Transformers (ViTs) that can obtain competitive performance in comparison to comparable non-interpretable methods. To our knowledge, ComFe is the first interpretable head and unlike other interpretable approaches can be readily applied to large-scale datasets such as ImageNet-1K. Additionally, ComFe provides improved robustness and outperforms previous interpretable approaches on key benchmark datasets while using a consistent set of hyperparameters and without finetuning the pretrained ViT backbone. With only global image labels and no segmentation or part annotations, ComFe can identify consistent component features within an image and determine which of these features are informative in making a prediction. Code is available at github.com/emannix/comfe-component-features.

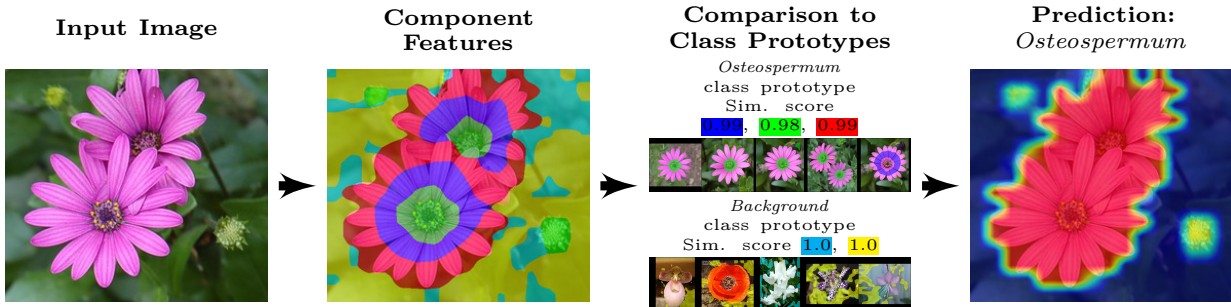

Figure 1: **Illustration of ComFe.** The image is first clustered into component features, which are then compared to class prototypes. This comparison between component features and class prototypes is used to identify the salient parts of the image for predicting the class *Osteospermum*, as shown by the class confidence heatmap in the final image.

# 1 Introduction

Deep learning is applied in numerous contexts to solve challenging computer vision problems, from identifying disease in medical imaging (Zhou et al., 2023), to species identification (Beloiu et al., 2023) and even in self-driving cars (Lee & Liu, 2023). However, standard deep learning approaches are *black boxes* (Rudin, 2019), and it can be challenging to determine whether a prediction is being made based on the most relevant features in an image. For example, neural networks can often learn spurious correlations from image backgrounds (Yang et al., 2022).

Post-hoc interpretability techniques, such as Class Activation Maps (CAM) (Zhou et al., 2016), can uncover this behaviour. However, these approaches can be misleading and cannot be used to understand which parts of the training dataset led to a particular classification (Rudin, 2019). Interpretable models that are designed to reason in a logical fashion, can achieve this goal by identifying prototypical parts within the training dataset that provide evidence for a particular category (Chen et al., 2019; Nauta et al., 2023; Xue et al., 2024).

Nevertheless, interpretable models can have a poor semantic correspondence between the embeddings of prototypes and their visual characteristics (Kim et al., 2022; Hoffmann et al., 2021). Current interpretable approaches are also less accessible in comparison to standard black-box approaches: they add new hyperparameters which need to be tuned for each dataset, and scale poorly to larger networks or datasets with a large number of classes. Current approaches do not report performance on large scale datasets, such as ImageNet-1K, and doing so *would require significant compute and hyperparameter tuning to produce meaningful results.* Foundation models trained on large datasets using self-supervised learning and a Vision Transformer (ViT) architecture, such as the DINOv2 family (Oquab et al., 2024), could help solve these challenges. These models produce embeddings across a range of vision contexts that faithfully reflects semantic similarity (Oquab et al., 2024), and a classifier that can explain predictions using this latent space has the potential to be easy to train, highly scalable and provide good semantic correspondence.

This paper presents **Com**ponent **Fe**atures (ComFe), a modular and efficient interpretable-by-design image classification head that addresses these challenges. To the best of our knowledge, ComFe is the first classifier head for downstream prediction that provides interpretability intrinsically, rather than being a post-hoc technique to explain a model's behaviour. ComFe is designed to be used with a pretrained ViT network, and employs a transformer decoder (Vaswani et al., 2017) architecture and a hierarchical mixture modelling framework to make explainable predictions. As shown in Fig. 1, ComFe identifies a set of component features within an image and compares them to a library of class prototypes learned from the training data which can be used to answer *why* a particular prediction is made. This approach is inspired by Detection Transformer (Carion et al., 2020), Mask2Former (Cheng et al., 2022) and PlainSeg (Hong et al., 2023), in addition to recent advances in semi-supervised learning (Assran et al., 2021; Mo et al., 2023; Mannix & Bondell, 2023). In this work, we:

- Present ComFe, the *first interpretable image classification head, and the first interpretable method that is sufficiently scalable to report results on the ImageNet-1K dataset.* ComFe achieves competitive results with comparable non-interpretable approaches, and provides improved performance on a range of ImageNet-1K generalisability and robustness benchmarks.

- *Demonstrate the competitive performance of ComFe* on a range of datasets using a consistent set of hyperparameters in comparison to other interpretable approaches that finetune the backbone networks and tune the hyperparameters for each dataset.

- Highlight how ComFe can *consistently detect particular regions* of an image (e.g. a bird's head, body and wings), identify informative versus non-informative (i.e. background) regions, and explain why particular predictions are made on the basis of similarity to class prototypes that are representative of the training data.

## 2 Related Work

**Interpretable computer vision models.** ProtoPNet (Chen et al., 2019) was the first method to show that deep learning based computer vision models could be designed to explain their predictions and obtain performance similar to non-interpretable approaches. They introduced prototypes for interpretability—visual representations of concepts from the training data that can be used to explain *why* a model makes a particular prediction. This allowed for classification to be based on the distance between image patch embeddings and these prototypes within the latent space (Chen et al., 2019). To ensure the interpretability of these prototypes, they constrained them to the embeddings of a patch within the training dataset.

A number of further works, including ProtoTree (Nauta et al., 2021), ProtoPShare (Rymarczyk et al., 2021), ProtoPool (Rymarczyk et al., 2022), ProtoPFormer (Xue et al., 2024) and PIP-Net (Nauta et al., 2023) are derived from this approach. ProtoTree incorporates a decision tree into the reasoning process, while the other approaches focus on addressing a key weakness of ProtoPNet—the learned prototypes are often background features in the training data, and these need to be pruned to obtain good interpretability. ProtoPShare, PIP-Net and ProtoPool take different approaches to achieve this, through encouraging prototypes to be shared between classes to improving their sparsity. ProtoPFormer improves the application of ProtoPNet to Vision Transformers (ViT) (Dosovitskiy et al., 2021; Touvron et al., 2021) and presents a new loss that further encourages learned prototypes to attend to foreground rather than background features.

In contrast, the INTR (Paul et al., 2024) approach uses the cross-attention layer from a transformer decoder head to produce explanations of image classifications. While elegant, this moves away from prototypes being associated with the training data, making these models less transparent than the ProtoP family. It is also noted that later works, such as PIP-Net, move away from the inflexibility of the prototypes being determined by the training data, and instead interpret and visualise prototypes by using the closest training example within the latent space, or the one with the highest probability under the model (Nauta et al., 2023).

Nevertheless, these methods can be challenging to adapt to new datasets as they introduce a number of additional hyperparameters to the training process. This is highlighted by most papers in this field only using three benchmarks—CUB200 (Wah et al., 2011), Oxford Pets (Parkhi et al., 2012) and Stanford Cars (Krause et al., 2013)—with hyperparameter tuning being undertaken for each dataset. They can also be computationally expensive to fit in comparison to non-interpretable approaches and require large amounts of GPU memory.

**Self-supervised learning and foundation models.** In computer vision, self-supervised approaches train neural networks as functional mappings between the image domain and a representation or latent space that captures the semantic information contained within the image with minimal loss (Chen et al., 2020a; 2021; 2020b). State-of-the-art self-supervised approaches can obtain competitive performance on downstream tasks such as image classification, segmentation and object detection when compared to fully supervised approaches without any further fine-tuning (Tukra et al., 2023; Oquab et al., 2024). This is achieved by using techniques such as bootstrapping (Grill et al., 2020), masked autoencoders (He et al., 2022), or contrastive learning (Chen et al., 2020a), which use image augmentations to design losses that encourage a neural network to project similar images into similar regions of the latent space. By combining these approaches on large scale datasets, ViT foundation models such as DINOv2 (Oquab et al., 2024) can be trained that perform well across a range of visual contexts. Contrastive language-image pretraining (CLIP) can also be an effective self-supervised approach, but it requires large image datasets with captions to train models (Radford et al., 2021; Cherti et al., 2023).

## 3 Methodology

**Motivation.** Underlying the reasoning process of the ProtoP family of interpretable approaches is the encoder network. If this network performs poorly, then the *explanation* provided by ProtoP approaches will be nonsensical (Hoffmann et al., 2021; Kim et al., 2022). Computer vision foundation models, like the DINOv2 family, provide highly informative patch features that produces an embedding space where the cosine similarity between image patches describes their degree of semantic similarity (Oquab et al., 2024). It has also

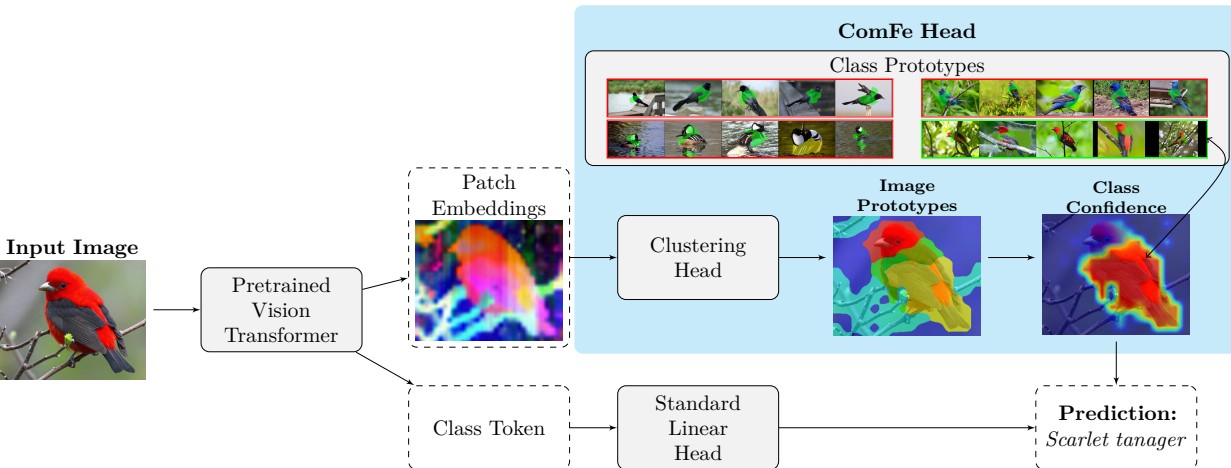

Figure 2: **The ComFe framework.** A pretrained ViT network is applied to an input image $\mathbf{X}$, to produce patch embeddings $\mathbf{Z}$ and a class token. The ComFe clustering head $g_\theta$ is trained to cluster patch embeddings and produce image prototypes $\mathbf{P}$, which are compared to a set of learnt class prototypes $\mathbf{C}$ which represent the training data to identify the informative image regions and make a prediction. In constrast, a standard non-interpretable approach will train a linear head to make a prediction using the class token, and does not provide insight into the informative image regions or which training images led to a particular prediction.

been found that particular directions within the latent space of these models will encode particular concepts (Bhalla et al., 2024). This work addresses the challenge of leveraging these foundation model features to build a modular interpretable image classification head that can be applied to a pretrained Vision Transformer (ViT) (Dosovitskiy et al., 2021) network.

**Interpretability with component features.** Similarly to the ProtoP methods, we aim to learn class prototypes $\mathbf{C}$, which are directions within this feature space that relate to a set of concepts of interest (e.g. species of birds). In particular, we want to learn class prototypes in the latent space of the output patch embeddings $\mathbf{Z}$ of the network, as this will allow the regions of an image that contribute to a particular classification to be identified. Further, these class prototypes need to be learnt in a way that represents the training data, so they can be used to answer *why* a particular prediction is made.

This can be posed as a clustering problem, where the goal is to use distributions centred on the class prototypes $\mathbf{C}$ to cluster the patch embeddings $\mathbf{Z}$ into meaningful groups. Under this framework, the closest training data patches to the centre of these clusters can function as *exemplars*, and can be used to visualise each class prototype.

However, the output of a ViT network can have hundreds of patches per image, many of which will be similar, particularly if they make up the same part of a particular object or scene. To reduce the complexity of solving the class prototype clustering problem, image prototypes $\mathbf{P}$ are introduced. These represent component features within an image, which can be used to consider a small set of image regions instead of a much larger number of image patches. To learn the image prototypes themselves, a second, nested clustering problem is introduced to learn a clustering head $g_\theta$, that generates the image prototypes parametrically on the basis of the patch embeddings $\mathbf{Z}$. This facilitates interpretable inference that is more efficient than considering individual patches, and this approach is illustrated in Fig. 1.

**Notation.** We refer to an RGB image as $\mathbf{X} \in \mathbb{R}^{3 \times h \times w}$, where $w$ is the image width and $h$ is the height. Throughout this paper, we use bolded characters $\mathbf{A}$ to refer to matrices, and the notation $\mathbf{A}_{i:}$ refers to the $i^{\text{th}}$ row of a matrix while $\mathbf{A}_{:j}$ refers to the $j^{\text{th}}$ column. The respective lowercase letter with two subscripts $a_{ij}$ refers to a specific element of the matrix $\mathbf{A}$. We consider a frozen pretrained ViT encoder model $f$ and a clustering head $g_\theta$, where $\theta$ refers to the parameters of this network.

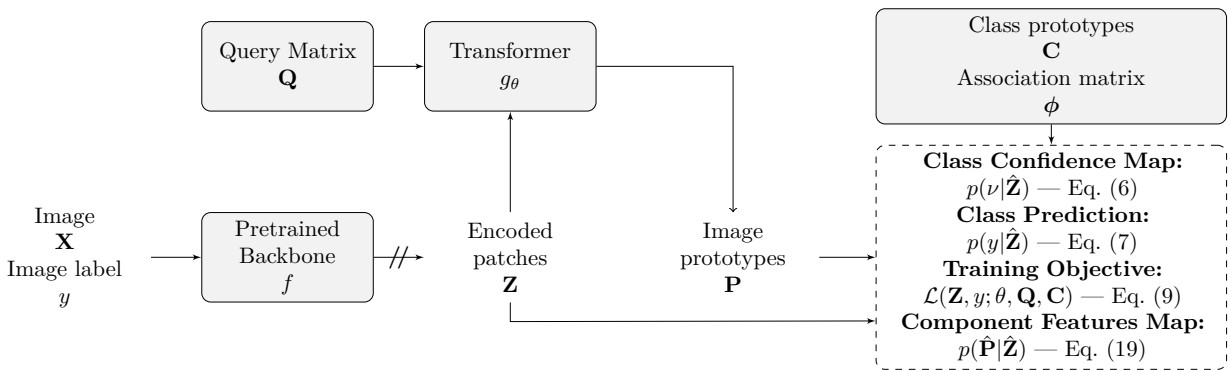

Figure 3: **Summary of the ComFe framework.** Given an input image $\mathbf{X}$, the patch embeddings $\mathbf{Z}$ are obtained using a pretrained ViT backbone model, $f(\mathbf{X}) = \mathbf{Z}$. These patches are clustered into component features using a set of image prototypes $\mathbf{P}$, that are obtained from a transformer decoder clustering head as described in Eq. (5). In producing a classification, the image prototypes $\mathbf{P}$ are compared to the class prototypes $\mathbf{C}$, as described in Eq. (6) and Eq. (7). The variable $\nu$ represents the class prediction of a local patch, which is unknown as segmentation labels are assumed to not be available. The component features map (image prototypes) are visualised based on the image prototype $\mathbf{P}$ with the greatest likelihood of generating a particular patch in Eq. (19).

**Modelling framework.** To formalise the model, we consider three key components: (1) the patch embeddings $\mathbf{Z} = f(\mathbf{X}) \in \mathbb{R}^{N_Z \times d}$, where $d$ is the dimensionality of the latent space and $N_Z$ is the number of patches created created by the pretrained model $f$; (2) the unknown patch class labels $\nu$, described by the class prototypes $\mathbf{C} \in \mathbb{R}^{N_C \times d}$, where we have $N_C$ class prototypes with a fixed association to each class as defined by a smoothed one-hot encoded matrix $\phi \in \mathbb{R}^{N_C \times c}$, with smoothing parameter $\alpha$; and (3) image prototypes $\mathbf{P} \in \mathbb{R}^{N_P \times d}$ which represent component features within an image, where $N_P$ is the number of desired image prototypes and is much smaller than the number of image patches $N_Z$.

Prior works (Caron et al., 2021; Ayzenberg et al., 2024) have found the $\ell^2$ normalised form $\hat{\mathbf{Z}}_{i:} = \frac{\mathbf{Z}_{i:}}{\|\mathbf{Z}_{i:}\|}$ of the outputs provide an effective measure of semantic similarity, through the cosine distance metric. This also facilitates fitting distributions in the high dimensional space of these embeddings (Sra, 2018). We consider the following joint probability based on a hierarchical mixture modelling approach (Marin et al., 2005; Zhou et al., 2020):

$$p(\hat{\mathbf{Z}}_{i:}, \hat{\mathbf{P}}_{j:}, \nu) := p(\hat{\mathbf{Z}}_{i:}|\hat{\mathbf{P}}_{j:})p(\hat{\mathbf{P}}_{j:}|\nu)p(\nu), \tag{1}$$

where $p(\nu)$ is the prior distribution for the patch classes in the dataset. This joint probability describes a generative model where the patch class $\nu$, parameterised by class prototypes $\mathbf{C}$, describes the distribution of the $j^{\text{th}}$ image prototype $\mathbf{P}_{j:}$, which in turn describes the distribution of the $i^{\text{th}}$ patch embedding $\mathbf{Z}_{i:}$. This defines the image patches $\mathbf{Z}$ as independent of the patch class $\nu$ when conditioned on an image prototype $\mathbf{P}_{j:}$. We define these distributions in terms of the von-Mises Fisher distribution on the unit sphere in $\mathbb{R}^d$ (denoted $\mathbb{S}^d$) (Govindarajan et al., 2023) with density

$$\text{VMF}_d(\boldsymbol{x}; \boldsymbol{\mu}, \tau) = C_d(\tau) \exp(\tau^{-1}\boldsymbol{x} \cdot \boldsymbol{\mu}), \quad \boldsymbol{x} \in \mathbb{S}^d, \tag{2}$$

with mean vector $\boldsymbol{\mu} \in \mathbb{S}^d$, concentration parameter $\tau$, and normalizing constant $C_d(\tau)$. The model becomes:

$$p(\hat{\mathbf{Z}}_{i:}|\hat{\mathbf{P}}_{j:}) := \text{VMF}_d(\hat{\mathbf{Z}}_{i:}; \hat{\mathbf{P}}_{j:}, \tau_1) \tag{3}$$

$$p(\hat{\mathbf{P}}_{j:}|\nu) := \sum_{l=1}^{N_C} \phi_{l\nu}\text{VMF}_d(\hat{\mathbf{P}}_{j:}; \hat{\mathbf{C}}_{l:}, \tau_2). \tag{4}$$

**Parameterising the prototypes.** The class prototypes $\mathbf{C}$ are parameterised as a matrix of learnable parameters, while the image prototypes $\mathbf{P}$ are fit parametrically following an amortised-inspired approach

(Margossian & Blei, 2024). To promote the efficiency of ComFe, we do not learn the full distribution of each $\hat{\mathbf{P}}_{j:}$ and instead use a point Maximum Likelihood Estimate (MLE). For each image, this is done by generating $\mathbf{P}_{j:}$ deterministically using a transformer decoder $g_\theta$ (Vaswani et al., 2017), which we refer to as the clustering head

$$\mathbf{P}_{i:}(\mathbf{Z}; \theta, \mathbf{Q}_{i:}) = g_\theta(\mathbf{Z}, \mathbf{Q}_{i:}), \tag{5}$$

where $\mathbf{Q} \in \mathbb{R}^{N_P \times d}$ is a learnable query matrix, that prompts the decoder to calculate the image prototypes by considering the entire patch representation of the image $\mathbf{Z}$. For clarity, we leave the arguments $\theta$, $\mathbf{Q}_{i:}$ and $\mathbf{Z}$ of $\mathbf{P}_{i:}(\cdot)$ as implied, and simply write $\mathbf{P}_{i:}$.

**Deriving a likelihood.** To predict the class that a patch belongs to, Bayes' rule is used and the image prototypes are marginalised out to obtain

$$p(\nu|\hat{\mathbf{Z}}_{i:}) \approx \sum_{j=1}^{N_P} \sigma(\hat{\mathbf{P}}_{j:} \cdot \hat{\mathbf{C}}/\tau_2) \phi_{:\nu} \sigma(\hat{\mathbf{Z}}_{i:} \cdot \hat{\mathbf{P}}/\tau_1)_j \tag{6}$$

where $\sigma(\cdot)$ is the softmax function and the approximation arises from using a MLE point estimate for $\hat{\mathbf{P}}_{j:}$. Uniform categorical prior distributions are assumed, and further details are given in Section A in the supporting information.

The main challenge here is that $\nu$ is unknown. We only assume access to image classification annotations $y$, which describe the global image label as a one-hot encoded class vector. To connect the unknown $\nu$ class of the image patches to the global annotation $y$, a vector of binary probabilities is constructed using max-pooling

$$p(y_l|\hat{\mathbf{Z}}) := \max_i p(\nu = l|\hat{\mathbf{Z}}_{i:}), \tag{7}$$

which describes the maximum probability given by a patch that class $l$ is present in the image. This provides the likelihood

$$-\log p(\hat{\mathbf{Z}}, y) = -\log p(\hat{\mathbf{Z}}) + \text{BCE}(y, p(y_l|\hat{\mathbf{Z}})), \tag{8}$$

where the first term describes how well the image prototypes are clustered, and $\text{BCE}(y, p(y_l|\hat{\mathbf{Z}})) = -\sum_{l=1}^c y_l \log p(y_l|\hat{\mathbf{Z}}) - \sum_{l=1}^c (1 - y_l) \log(1 - p(y_l|\hat{\mathbf{Z}}))$ denotes the binary cross-entropy loss for a multi-label classification problem.

**Training objective.** Based on the likelihood above, the training objective for ComFe is given by

$$\mathcal{L}(\mathbf{Z}, y; \theta, \mathbf{Q}, \mathbf{C}) = \mathcal{L}_{\text{discrim}}(\mathbf{Z}, y; \theta, \mathbf{Q}, \mathbf{C}) \tag{9}$$
$$+ \mathcal{L}_{\text{cluster}}(\mathbf{Z}; \theta, \mathbf{Q}) + \mathcal{L}_{\text{aux}}(\mathbf{Z}, y; \theta, \mathbf{Q}, \mathbf{C}),$$

which is separated into the discriminative term $\mathcal{L}_{\text{discrim}}$ that uses a binary cross entropy loss and a clustering term $\mathcal{L}_{\text{cluster}}$. Additional auxiliary objectives $\mathcal{L}_{\text{aux}}$ are also specified which ensure the consistency of image prototypes, the uniqueness of class prototypes and improve the fitting process. The discriminative and clustering term are given by

$$\mathcal{L}_{\text{cluster}}(\mathbf{Z}; \theta, \mathbf{Q}) = -\frac{1}{N_Z} \sum_{i=1}^{N_Z} \log \sum_{j=1}^{N_P} \text{VMF}_d(\hat{\mathbf{Z}}_{i:}, \hat{\mathbf{P}}_{j:}; \tau_1) \tag{10}$$

$$\mathcal{L}_{\text{discrim}}(\mathbf{Z}, y; \theta, \mathbf{Q}, \mathbf{C}) = \text{BCE}(y, p(y_l|\hat{\mathbf{Z}})). \tag{11}$$

A more detailed description of the derivation and motivation of the clustering, discriminative and auxiliary terms is provided in Section A in the supporting information.

**Class prototype exemplars.** The class prototypes $\mathbf{C}$ represent the centres of von-Mises Fisher distributions. The image prototypes $\mathbf{P}$ from the training dataset with greatest cosine similarity to each class prototype can be used as representatives, to visualise the *concept* each one describes. These particular image prototypes are called class prototype exemplars, and are used to visualise the class prototypes in Fig. 1 and Fig. 2. Further examples are also shown in Fig. 4, and Fig. S5 and Fig. S6 in the supporting information. This reflects Eq. (6), where the cosine similarity of an image prototype to each class prototypes is used in the term

$$\text{Sim}(\mathbf{P}_{j:}) = \sigma(\hat{\mathbf{P}}_{j:} \cdot \hat{\mathbf{C}}/\tau_2) \tag{12}$$

for classifying an image. This measure in Eq. (12) was used to calculate the similarity values presented in Fig. 1 and elsewhere within the paper. These exemplars are only used to interpret the class prototypes post-training, similar to the prototypes in PIP-Net (Nauta et al., 2023), and are not used during the training process.

**Background classes.** To allow the model to learn which regions of an image are informative, an additional background class is added and it is assumed that each image has one or more background patches. This turns the multi-class image classification problem into a multi-label one, where the background class is added to the label for each image. Further, $N_N$ additional background class prototypes are added to $\mathbf{C}$ and $\phi$. This can be done without changing the form of the discriminative loss terms in Eq. (10) as the binary cross-entropy loss naturally extends to multi-label classification problems. Further details are provided in Section A in the supporting information.

**Comparison to closely related work.** The ComFe head shares a similar architecture to the INTR (Paul et al., 2024) approach, in that they both use a transformer decoder on the outputs of a backbone network to make a prediction. However, INTR uses the query $\mathbf{Q}$ to define classes — using one query vector per class — and relies on cross-attention within the decoder heads to interpret why the model is making a particular prediction. For ComFe, the query has a fixed size that depends on the number of image prototypes $N_P$, and the similarity of the image $\mathbf{P}$ to the class $\mathbf{C}$ prototypes is used to explain the model outputs. This makes ComFe much more scalable to datasets such as ImageNet, as the number of image prototypes $N_P$ is much smaller than the number of classes. This allows for models to be easily trained with large batch sizes on a single GPU.

Recently, Zhu et al. (2025) introduced a new ProtoP inspired method, which we refer to as ProtoNonParam, designed for use with foundation models. ProtoNonParam consists of three key steps; (i) foreground extraction, using a PCA based approach (Oquab et al., 2024) to identify the objects of interest for classification, (ii) non-parametric prototype learning, where the challenge of identifying foreground patches of a known class with part-prototypes is posed as an optimal transport problem and solved using the Sinkhorn-Knopp algorithm, and (iii) feature space finetuning. In this last step, the part-prototype vectors and foundation model weights are frozen, and additional transformer encoder blocks are interleaved through the original layers and finetuned (Zhu et al., 2025). While this approach is able to effectively exclude backgrounds features from becoming part prototypes through the foreground extraction step, this limits the flexibility of the approach in cases where there are multiple possible foreground objects present.

## 4 Experiments

### 4.1 Implementation details

The weights of the pretrained ViT network $f$ are frozen during training, and the parameters of the clustering head $g_\theta$ are randomly initialised using a Xavier normal distribution (Glorot & Bengio, 2010). The class prototypes $\mathbf{C}$ are randomly initialised from a uniform distribution on the unit sphere, and the initial queries $\mathbf{Q}$ are sampled from a standard normal distribution. Standard choices for training transformers are used, such as the AdamW optimiser (Loshchilov & Hutter, 2017b), cosine learning rate decay with linear warmup (Loshchilov & Hutter, 2017a; Gotmare et al., 2019) and gradient clipping (Pascanu et al., 2013). For image augmentations, we follow DINOv2 and other works, including random cropping, flipping, color distortion and random greyscale (Chen et al., 2020a; Oquab et al., 2024). For the architecture of the clustering head $g_\theta$, two

transformer decoder layers with eight attention heads are used, and following PlainSeg (Hong et al., 2023) the loss is calculated after each decoder layer and averaged.

For each dataset, a total of five image prototypes $\mathbf{P}$—giving $\mathbf{Q}$ five rows—and $6c$ class prototypes $\mathbf{C}$ are used, where $c$ is the number of classes in the dataset. The first $3c$ class prototypes are assigned to each class (for three per class), while the remaining $3c$ are assigned to the background class. Similar temperature parameters are used to previous works (Assran et al., 2021), with $\tau_1 = 0.1$, $\tau_2 = 0.02$ and $\tau_c = 0.02$. An ablation study on these hyperparameters is undertaken in Section D of the supporting information.

The same set of hyperparameters are used across all of the training runs, with the exception of the batch size which is increased from 64 image to 1024 for only the ImageNet dataset. The number of epochs for ImageNet (Russakovsky et al., 2015) is also reduced, from 50 epochs per training run to 20 epochs. All results use an input resolution of $224 \times 224$ pixels, with outputs upsampled using bilinear interpolation to generate pixel-level results from patch-level predictions.

## 4.2 Datasets

Finegrained image benchmarking datasets including Oxford Pets (37 classes) (Parkhi et al., 2012), FGVC Aircraft (100 classes) (Maji et al., 2013), Stanford Cars (196 classes) (Krause et al., 2013) and CUB200 (200 classes) (Wah et al., 2011) have all previously been used to benchmark interpretable computer vision models. These are used to test the performance of ComFe, in addition to other datasets including ImageNet-1K (1000 classes) (Russakovsky et al., 2015), CIFAR-10 (10 classes), CIFAR-100 (100 classes) (Krizhevsky et al., 2009), Flowers-102 (102 classes) (Nilsback & Zisserman, 2008) and Food-101 (101 classes) (Bossard et al., 2014). These cover a range of different domains and have been previously used to evaluate the linear fine-tuning performance of the DINOv2 models (Oquab et al., 2024).

Further test datasets are considered for ImageNet that are designed to measure model generalisability and robustness. ImageNet-V2 (Recht et al., 2019) follows the original data distribution but contains harder examples, Sketch (Wang et al., 2019) tests if models generalise to sketches of the ImageNet class and ImageNet-R (Hendrycks et al., 2021a) tests if models generalise to art, cartoons, graphics and other renditions. Finally, the ImageNet-A (Hendrycks et al., 2021b) test dataset consists of real-world, unmodified and naturally occurring images of the ImageNet classes that are often misclassified by ResNet models. While ImageNet-V2 and Sketch contain all of the ImageNet classes, ImageNet-A and ImageNet-R only provide images for a 200 class subset.

## 4.3 Main results

**ComFe obtains competitive performance in comparison to previous interpretable approaches.** Table 1 shows that ComFe heads obtain competitive performance with previous interpretable models of similar sizes. We note that in some cases this is an imperfect comparison, with ComFe using the strong DINOv2 backbones, and prior works using networks initialized with ImageNet weights. We further include the benchmarks published by Zhu et al. (2025), that train interpretable models with the DINOv2 backbone by adding additional transformer blocks, which are finetuned while the original weights are kept frozen. When these additional blocks are finetuned, methods such as TesNet (Wang et al., 2021), EvalProtoPNet Huang et al. (2023) and ProtoNonParam can slightly outperform ComFe on CUB200, while ComFe still performs strongly in comparison on Stanford Cars. Without this finetuning step, the ProtoNonParam (Zhu et al., 2025) approach performs very poorly in comparison to ComFe.

To our knowledge ComFe is the first interpretable head, and while not all benchmarks in Table 1 are directly comparable, we present them to highlight the utility of this approach. We also note that where ComFe uses the same set of hyperparameters for all datasets in Table 1, other works undertake hyperparameter tuning for each task (Paul et al., 2024; Nauta et al., 2023; Xue et al., 2024).

**ComFe obtains competitive performance compared to a non-interpretable linear head.** Table 2 shows that ComFe is competitive with a linear head, the standard approach of training non-interpretable models with a frozen backbone. ComFe achieves improved performance on a number of datasets, including

Table 1: **Interpretable performance comparison.** Performance (top-1 accuracy) of ComFe and other interpretable image classification approaches on several benchmarking datasets. INTR and ComFe use transformer decoder heads, which adds additional model parameters (#P), while the benchmarks reported by Zhu et al. (2025) introduce $K$ additional transformer blocks to the DINOv2 backbone to support feature space finetuning.

| | Method | | Backbone | | Dataset | | | |
|---|---|---|---|---|---|---|---|---|
| | | #P | | #P | CUB | Pets | Cars | Aircr |
| **Interpretable** | ProtoPNet (Chen et al., 2019) | | ResNet-34 | 22M | 79.2 | | 86.1 | |
| (*Finetuned backbones*) | ProtoTree (Nauta et al., 2021) | | ResNet-34 | 22M | 82.2 | | 86.6 | |
| | ProtoPShare (Rymarczyk et al., 2021) | | ResNet-34 | 22M | 74.7 | | 86.4 | |
| | ProtoPool (Rymarczyk et al., 2022) | | ResNet-50 | 26M | 85.5 | | 88.9 | |
| | ProtoPFormer (Xue et al., 2024) | | CaiT-XXS-24 | 12M | 84.9 | | 90.9 | |
| | ProtoPFormer (Xue et al., 2024) | | DeiT-S | 22M | 84.8 | | 91.0 | |
| | PIP-Net (Nauta et al., 2023) | | ConvNeXt-tiny | 29M | 84.3 | 92.0 | 88.2 | |
| | PIP-Net (Nauta et al., 2023) | | ResNet-50 | 26M | 82.0 | 88.5 | 86.5 | |
| | INTR (Paul et al., 2024) | 10M | ResNet-50 | 26M | 71.8 | 90.4 | 86.8 | 76.1 |
| | ProtoPNet (Zhu et al., 2025) | | DINOv2 ViT-S/14 ($K = 5$) | 21M+9M | 85.5 | | 79.5 | |
| | TesNet (Zhu et al., 2025) | | DINOv2 ViT-S/14 ($K = 5$) | 21M+9M | **89.0** | | 82.6 | |
| | EvalProtoPNet (Zhu et al., 2025) | | DINOv2 ViT-S/14 ($K = 5$) | 21M+9M | 88.2 | | 87.1 | |
| | ProtoNonParam (Zhu et al., 2025) | | DINOv2 ViT-S/14 ($K = 5$) | 21M+9M | 88.2 | | 86.1 | |
| (*Frozen backbones*) | ComFe | 8M | DINOv2 ViT-S/14 (f) | 21M | 87.6 | **94.6** | **91.1** | **77.5** |
| | ProtoNonParam* (Zhu et al., 2025) | | DINOv2 ViT-S/14 (f) | 21M | 78.0 | | | |

Table 2: **Non-interpretable performance comparison.** Performance (top-1 accuracy) of ComFe versus a linear head (Oquab et al., 2024) with frozen features on several benchmarking datasets.

| Method | | Backbone | | Dataset | | | | | | | | |
|---|---|---|---|---|---|---|---|---|---|---|---|---|
| | #P | *DINOv2* | #P | IN-1K | C10 | C100 | Food | CUB | Pets | Cars | Aircr | Flowers |
| Linear (Oquab et al., 2024) | <1M | ViT-S/14 (f) | 21M | 81.1 | 97.7 | 87.5 | 89.1 | 88.1 | 95.1 | 81.6 | 74.0 | 99.6 |
| Linear (Oquab et al., 2024) | <1M | ViT-B/14 (f) | 86M | 84.5 | 98.7 | 91.3 | 92.8 | 89.6 | 96.2 | 88.2 | 79.4 | 99.6 |
| Linear (Oquab et al., 2024) | <1M | ViT-L/14 (f) | 300M | 86.3 | 99.3 | 93.4 | 94.3 | **90.5** | **96.6** | 90.1 | 81.5 | **99.7** |
| ComFe | 8M | ViT-S/14 (f) | 21M | 83.0 | 98.3 | 89.2 | 92.1 | 87.6 | 94.6 | 91.1 | 77.5 | 99.0 |
| ComFe | 32M | ViT-B/14 (f) | 86M | 85.6 | 99.1 | 92.2 | 94.2 | 88.3 | 95.3 | 92.6 | 81.1 | 99.3 |
| ComFe | 57M | ViT-L/14 (f) | 300M | **86.7** | **99.4** | **93.6** | **94.6** | 89.2 | 95.9 | **93.6** | **83.9** | 99.4 |

ImageNet, CIFAR-10, CIFAR-100, Food-101, StanfordCars and FGVC Aircraft. For smaller models, we observe that ComFe outperforms the linear head by a larger margin. For species identification datasets, such as CUB200, Oxford Pets and Flowers-102, ComFe results in slightly lower performance compared to the linear head. Further investigations into the CUB200 dataset suggest that this may be due to incorrect annotations in the training data, as explored in Section B.

Table 3: **Generalisation and robustness benchmarks.** Performance (top-1 accuracy) of ComFe versus a linear head with frozen features on ImageNet-1K generalisation and robustness benchmarks. For IN-R and IN-A accuracy is reported across all classes rather than restricting to the subset of classes included in these datasets. The accuracy only considering these classes is provided in Table S14.

| Method | | Backbone | | Test Dataset | | | |
|---|---|---|---|---|---|---|---|
| | #P | *DINOv2* | #P | IN-V2 | Sketch | IN-R | IN-A |
| Linear (Oquab et al., 2024) | <1M | ViT-S/14 (f) | 21M | 70.9 | 41.2 | 37.5 | 18.9 |
| Linear (Oquab et al., 2024) | <1M | ViT-B/14 (f) | 86M | 75.1 | 50.6 | 47.3 | 37.3 |
| Linear (Oquab et al., 2024) | <1M | ViT-L/14 (f) | 300M | 78.0 | 59.3 | 57.9 | 52.0 |
| ComFe | 8M | ViT-S/14 (f) | 21M | 73.4 | 45.3 | 42.1 | 26.4 |
| ComFe | 32M | ViT-B/14 (f) | 86M | 77.3 | 54.8 | 51.9 | 43.2 |
| ComFe | 57M | ViT-L/14 (f) | 300M | **78.7** | **59.5** | **58.8** | **55.0** |

**ComFe improves generalisability and robustness.** Table 3 shows that ComFe is able to improve performance on the ImageNet-V2 test set while also improving on generalisability and robustness benchmarks

Table 4: **Training efficiency**. Comparison of timings and GPU memory requirements for interpretable approaches trained on the CUB200 dataset using an NVIDIA 80GB A100 GPU and a batch size of 64.

| Method | Backbone | | Epoch time | GPU mem | Epochs |
|---|---|---|---|---|---|
| ProtoPFormer | ViT-L/16 | 300M | 42s | 28GB | 200 |
| PIP-Net | ConvNeXt-tiny | 29M | 121s | 42GB | 60 |
| ComFe | 57M DINOv2 ViT-L/14 (f) | 300M | **34s** | **5GB** | **50** |

in comparison to a linear head. Previous work has shown that while self-supervised learning approaches such as CLIP can improve their performance on ImageNet and ImageNet-V2 by fine-tuning the model backbone, doing so can reduce performance on Sketch, ImageNet-A and ImageNet-R (Radford et al., 2021). This suggests that ComFe may be a viable option to improve model performance with a pretrained ViT network, without compromising on the generalisability and robustness of the model.

**ComFe is efficient.** Table 4 shows that ComFe is efficient to train in comparison to other interpretable approaches. For the CUB200 dataset, ComFe can train a ViT-L/14 model in thirty minutes on one 80GB NVIDIA A100 GPU, and has a sufficiently small memory footprint that it could be trained on most mid-range consumer graphics cards.

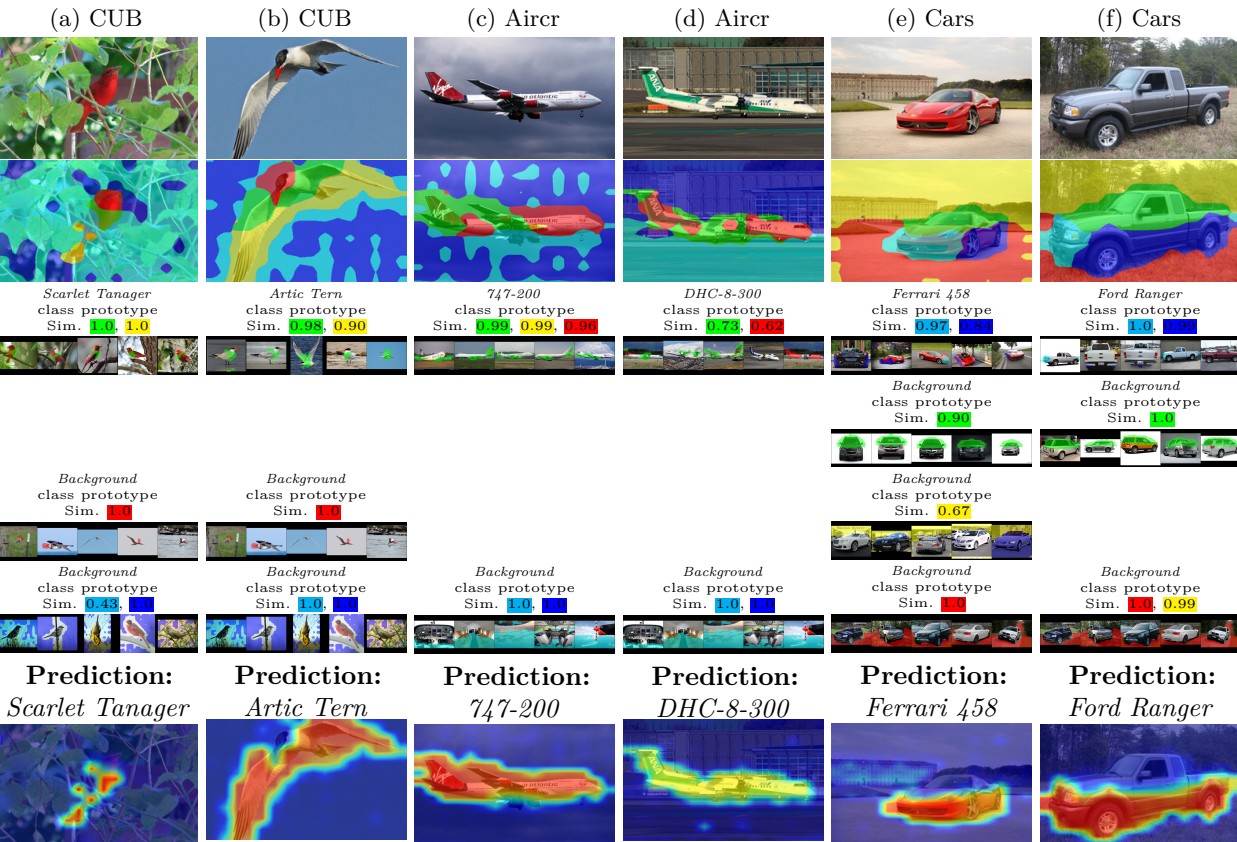

Figure 4: **Visualizing ComFe explanations.** Example ComFe predictions showcasing explainability, sampled from the validation images from the FGVC Aircraft, Stanford Cars and CUB200 datasets. The rows show the input images, component features (image prototypes), class prototype similarity and exemplars, and the class confidence heatmap for the final classification.

**The image prototypes learnt by ComFe identify visual components across classes.** In Fig. 4 it can be seen that the image prototypes **P** partition images containing different classes in a consistent manner.

On the CUB200 dataset, the first image prototype (red) identifies the head of birds, while the second image prototype (yellow) identifies the wings and tail feathers. The third image prototype (green) identifies the body, and the final two image prototypes (blue and cyan) are always associated with the background. Similar patterns are observed across the other datasets, with particular image prototypes identifying the front, side and upper cabin and roof of the car for Stanford Cars, and image prototypes locating the wings, tail and the body of a plane for the FGVC Aircraft dataset.

**The class prototypes learned by ComFe identify the category which image prototypes belong to.** Fig. 4 also shows how the image prototypes **P** are classified as informative (belonging to a specific class) or non-informative by the class prototypes **C**. We observe that when identifying birds the head appears to be non-informative on the CUB200 dataset, which reflects that a background class prototype has been found by ComFe on the ViT-S DINOv2 backbone that relates to a bird's head across different species. A similar phenomenon occurs in Stanford Cars, where two background class prototypes have been found that describe the upper vehicle cabin and roof of particular types of cars. Further visualisations of these class prototypes are shown in Fig. S5 in the supporting information.

Like other clustering algorithms (Al-Daoud & Roberts, 1996), ComFe is sensitive to the initialisation of the algorithm, and different features may be found to be informative with different seeds. This is further addressed in the discussion and Section C in the supporting information, where the features found to be informative can also be seen to reflect the chosen pretrained backbone Fig. S14.

Table 5: **Quantifying interpretabilty using FunnyBirds.** Performance of methods trained on FunnyBirds (Hesse et al., 2023) under accuracy, correctness, completeness, distractability and contrastivity and background independence measures, and example attribution maps.

| Method | Backbone | Measure | | | | | |
|---|---|---|---|---|---|---|---|
| | | Acc. | Cor. | Com. | Dist. | Con. | B.I. |
| ProtoPNet (Hesse et al., 2023) | ResNet-50 | 0.94 | 0.24 | 0.92 | 0.58 | 0.46 | 1.00 |
| ComFe | ViT-B/14 w/reg | 0.98 | 0.74 | 0.86 | 0.80 | 0.61 | 0.98 |
| (DINOv2) | ViT-S/14 w/reg | 0.97 | 0.67 | 0.88 | 0.60 | 0.54 | 0.96 |
| | ViT-B/14 | 0.97 | 0.57 | 0.01 | 1.00 | 0.51 | 0.98 |
| | ViT-S/14 | 0.96 | 0.49 | 0.96 | 0.73 | 0.51 | 0.98 |

ProtoPNet ResNet-50    ComFe ViT-B/14 w/reg    ComFe ViT-S/14 w/reg    ComFe ViT-B/14    ComFe ViT-S/14

**ComFe can be used to evaluate the interpretability of a pretrained ViT network.** We employ the FunnyBirds framework (Hesse et al., 2023) to quantify the interpretability of ViT backbones using a ComFe head. FunnyBirds builds a dataset of cartoonish birds, which are constructed from various discrete parts. This allows the framework to define and calculate a number of different measures directly, such as correctness (correlation between model outputs and the reported importance of particular regions), contrastivity (degree to which explanations are different for similar classes) and completeness (degree to which an explanation describes all aspects of the model's decision), which trades off against distractability (degree to which an explanation ignores irrelevant features in the image) (Hesse et al., 2023). Classification accuracy and background independence (the sensitivity of the output to removal of background objects) are also reported, and all scores range from 0-1 where 1 is perfect performance. To provide context, we compare the results of ComFe heads to ProtoPNet, noting that this comparison uses different backbones and metric interfaces. The FunnyBirds measures are more indicative than comprehensive, given the limitations of the artificial dataset, particularly when comparing across different types of explanations (Hesse et al., 2023).

Table 5 shows the results for ProtoPNet and ComFe heads using DINOv2 backbones with and without registers. ViT models larger than ViT-S/14 without registers have been observed to learn artefact patches—background patches that store global information—whereas the introduction of registers mitigates this effect and improves performance in larger models (Darcet et al., 2024). ComFe heads on frozen DINOv2 backbones provide stronger results than ProtoPNet on accuracy, correctness, contrastivity and distractability. With the exception of the DINOv2 ViT-B/14 model, competitive completeness is also achieved. Background independence is also similar between all of the models. While ViT patch embeddings can contain global information, when they are trained suitably, like the small DINOv2 foundation models and variants with registers (Darcet et al., 2024), the kernels that are learnt generally preserve local information. This is supported by the strong performance of a ComFe head with the DINOv2 ViT-B/14 w/reg backbone across these interpretability measures.

However, this is not the case for all ViT networks. The figures below Table 5 shows that the ComFe head for the DINOv2 ViT-B/14 network learns to classify using artefact patches—background patches contaminated with global information (Darcet et al., 2024). This is highlighted by the abysmal completeness score for this model in Table 5, which by finding no parts to be informative allows for a perfect distractability score. Contrastivity and correctness are more similar between the DINOv2 ViT-S/14 and ViT-B/14 models, as all bird parts (eyes, beak, legs, tail and wings) are given mostly equal weight—maximum confidence, or very little at all. This is not the case for the DINOv2 variants with registers, where the ComFe heads in the example images both identify that the eyes are not informative.

Other interpretability metrics designed for ProtoP approaches, such as the purity measure introduced with PIP-Net (Nauta et al., 2023), are based on the definition of ProtoP prototypes. This is not directly applicable to ComFe, as the measures would need to be reworked for image and class prototypes, preventing clear comparisons. Further results for ComFe using DINOv2 backbones with registers and other pretrained ViT models are included in Section C.

## 5 Discussion

**Comparison to ProtoP approaches.** The ProtoP family considers the relationship between prototypes and classes to be represented by a learnable linear layer, whose weights describe the relationship between prototypes and classes (Chen et al., 2019; Nauta et al., 2023). This layer is key to making the final class prediction, as inference is generally done by max-pooling the prototype activations and passing them through this matrix. The ComFe approach deviates from this by using two sets of prototypes, image prototypes $\mathbf{P}$ that describe particular components of an image, and class prototypes $\mathbf{C}$ which represent directions in the latent space that describe a particular concept (e.g. a type of car). In a sense, the class prototypes of ComFe are similar to the prototypes considered by ProtoPNet and related approaches, as they represent key directions within the embedding space that describe particular concepts. However, in ComFe these are preassigned to classes, and the similarity between these class prototypes and the image prototypes for a particular image determines the predicted class of an image.

**Limitations.** ComFe is designed to be used in conjunction with a high quality foundation model backbone. While these models, like DINOv2 (Oquab et al., 2024), can perform well in many contexts, they can still perform poorly on images that are very different from their training set (Zhang et al., 2024). As ComFe is not designed to fine-tune the foundation model backbone, it may struggle to perform well in these settings. However, it is expected that in future these vision foundation models will continue to improve in performance, as they are used as visual encoders in multimodal LLMs, such as LLaVA (Liu et al., 2023) and GPT-4 (OpenAI et al., 2024). For example, it has been recently shown that foundation models can be improved through combining CLIP pre-training objectives with dense self-supervised tasks, such as those used in DINOv2 (Tschannen et al., 2025).

It is further noted that ComFe may identify different features as informative within an image, depending on the algorithm's initialisation. Such variability is common in clustering methods (Al-Daoud & Roberts, 1996) and is explored further in Section C of the supporting information. In practice, this variability can be addressed by training multiple heads, whose outputs can either be combined in an ensemble or evaluated individually to select the best-performing head (Jain, 2010). In our experiments (Fig. S11 and Fig. S12), the

best-performing head typically yields a semantically meaningful confidence map. Ensembling is not explored further, as interpreting the results would be more complex. The features deemed informative also vary across backbones, reflecting differences in their embedding spaces (Fig. S14).

ComFe, as presented in this work, is designed for image-level classification, where global labels guide the learning of informative regions. As a result, not all parts of an object may be deemed informative, and some regions can be classified as background, leading to incomplete confidence maps. Extending ComFe to semantic segmentation, where supervision is provided by segmentation masks, could address this limitation by encouraging prototypes to capture the full extent of an object. Related approaches, such as ProtoNonParam (Zhu et al., 2025), employ foreground extraction to obtain more spatially complete confidence maps, and a similar strategy could also be used with ComFe if required.

## 6    Conclusion

This work presents ComFe, a modular interpretable-by-design image classification head that uses a transformer decoder architecture and a hierarchical mixture modelling approach to build classification models using a pretrained ViT backbone. ComFe learns to identify directions within the latent space defined by this pretrained network that relate to particular concepts. This allows for the model to explain how an image is classified, by examining the similarity between component features of the image and these class prototypes. ComFe is fast to train and has low GPU memory requirements, in addition to providing improved performance across a range of benchmarking datasets in comparison to other interpretable methods.

## Acknowledgements

We acknowledge the Traditional Custodians of the unceded land on which the research detailed in this paper was undertaken, the Wurundjeri Woi Wurrung and Bunurong peoples of the Kulin nation, and pay our respects to their Elders past and present. This research was undertaken using the LIEF HPC-GPGPU Facility hosted at the University of Melbourne. This Facility was established with the assistance of LIEF Grant LE170100200. This research was also undertaken with the assistance of resources and services from the National Computational Infrastructure (NCI), which is supported by the Australian Government. Evelyn J. Mannix was supported by an Australian Government Research Training Program Scholarship to complete this work.

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

# Supplementary Material

## A  Further derivations for the learning objective

**Predicting patch class membership.**  Bayes' rule can be used to obtain the probability that a patch belongs to a particular patch class $\nu$ under this model, assuming that the prior distributions for the image prototypes $p(j)$ and the patch classes $p(\nu)$ are uniform categorical distributions. This assumes, in the first instance, that each patch is equally likely to be generated by each image prototype, and each image prototype is equally likely to be generated by each patch class. First, the probability that an image prototype $\mathbf{P}_{j:}$ is generated by a particular patch class $\nu$ can be computed using Bayes rule, where Eq. (4) can be inserted to obtain

$$p(\nu|\hat{\mathbf{P}}_{j:}) = \frac{p(\hat{\mathbf{P}}_{j:}|\nu)p(\nu)}{p(\hat{\mathbf{P}}_{j:})} \tag{13}$$

$$= \frac{p(\hat{\mathbf{P}}_{j:}|\nu)p(\nu)}{\sum_l^c p(\hat{\mathbf{P}}_{j:}|\nu_l)p(\nu_l)} \tag{14}$$

$$= \sigma\left(\hat{\mathbf{P}}_{j:} \cdot \hat{\mathbf{C}}/\tau_2\right)\phi_{:\nu}, \tag{15}$$

where $\sigma(\cdot)$ is the softmax function, and $\hat{\mathbf{P}}_{j:}$ is the L2 normalised form of $\mathbf{P}_{j:}$.

Then, consider the probability that a patch $\mathbf{Z}_{i:}$ was generated by a specific image prototype $\mathbf{P}_{j:}$, where Eq. (3) can be inserted to obtain

$$p(\hat{\mathbf{P}}_{j:}|\hat{\mathbf{Z}}_{i:}) = \frac{p(\hat{\mathbf{Z}}_{i:}|\hat{\mathbf{P}}_{j:})p(\hat{\mathbf{P}}_{j:})}{p(\hat{\mathbf{Z}}_{i:})} \tag{16}$$

$$= \frac{p(\hat{\mathbf{Z}}_{i:}|\hat{\mathbf{P}}_{j:})p(\hat{\mathbf{P}}_{j:})}{\sum_{j=1}^{N_P} p(\hat{\mathbf{Z}}_{i:}|\hat{\mathbf{P}}_{j:})p(\hat{\mathbf{P}}_{j:})} \tag{17}$$

$$\approx \frac{p(\hat{\mathbf{Z}}_{i:}|\hat{\mathbf{P}}_{j:})}{\sum_{j=1}^{N_P} p(\hat{\mathbf{Z}}_{i:}|\hat{\mathbf{P}}_{j:})} \tag{18}$$

$$= \sigma\left(\hat{\mathbf{Z}}_{i:} \cdot \hat{\mathbf{P}}/\tau_1\right)_j. \tag{19}$$

To facilitate the calculation of $p(\hat{\mathbf{P}}_{j:}|\hat{\mathbf{Z}}_{i:})$ we treat $p(\hat{\mathbf{P}}_{j:})$ as prior with a uniform distribution over the sphere to obtain the approximation in Eq. (18).

This gives the probability that a patch $\mathbf{Z}_{i:}$ belongs to a particular patch class $\nu$, through

$$p(\nu \mid \hat{\mathbf{Z}}_{i:}) = \sum_{j=1}^{N_P} \int_{\hat{\rho}_{j:} \in \mathbb{S}^d} p(\nu \mid \hat{\rho}_{j:}) p(j) \, p(\hat{\rho}_{j:} \mid \hat{\mathbf{Z}}_{i:}) \, d\hat{\rho}_{j:} \tag{20}$$

$$\approx \sum_{j=1}^{N_P} p(\nu \mid \hat{\mathbf{P}}_{j:}) \, p(\hat{\mathbf{P}}_{j:} \mid \hat{\mathbf{Z}}_{i:}), \tag{21}$$

where we approximate the intractable integral by using a point estimate, inserting the $\hat{\rho}_{j:} = \hat{\mathbf{P}}_{j:}$ direction that maximises the likelihood of the model, generated using Eq. (5) while optimising for $\theta$. The image prototypes $\mathbf{P}_{j:}$ are treated like a latent variable relating a patch $\mathbf{Z}_{i:}$ to its patch class $\nu$, that are marginalised from the expression.

**Deriving the likelihood.**  We do not have data to directly train a model on the patch classes $\nu$, as they are unknown. Instead, we use the global image class labels $y$ as a one-hot encoded class vector. The probability

of a class being present within an image can be defined as the maximum class probability over all image patches, creating a vector of Bernoulli distributions, of which the $l^{\text{th}}$ element is given by

$$p(y_l|\hat{\mathbf{Z}}) :\, = \max_i p(\nu_l|\hat{\mathbf{Z}}_{i:}). \tag{22}$$

Using this approach, the negative log likelihood of the model for a single image is

$$-\log p(\hat{\mathbf{Z}}, y) = -\log p(\hat{\mathbf{Z}}) - \left( \sum_{l=1}^{c} y_l \log p(y_l|\hat{\mathbf{Z}}) + (1 - y_l) \log \left( 1 - p(y_l|\hat{\mathbf{Z}}) \right) \right), \tag{23}$$

where the discriminative portion of the loss is given by the following binary cross entropy loss

$$\mathcal{L}_{\text{discrim}}(\mathbf{Z}, y; \theta, \mathbf{Q}, \mathbf{C}) = -\left( \sum_{l=1}^{c} y_l \log p(y_l|\hat{\mathbf{Z}}) + (1 - y_l) \log(1 - p(y_l|\hat{\mathbf{Z}})) \right), \tag{24}$$

and the clustering portion of the loss is given by the marginal distribution

$$\log p(\hat{\mathbf{Z}}) = \frac{1}{N_Z} \sum_{i=1}^{N_Z} \log p(\hat{\mathbf{Z}}_{i:}) \tag{25}$$

$$= \frac{1}{N_Z} \sum_{i=1}^{N_Z} \log \sum_{j=1}^{N_P} \int_{\hat{\rho}_{j:} \in \mathbb{S}^d} p(\hat{\mathbf{Z}}_{i:}|\hat{\rho}_{j:})p(j)p(\hat{\rho}_{j:}) \, d\hat{\rho}_{j:} \tag{26}$$

$$\approx \frac{1}{N_Z} \sum_{i=1}^{N_Z} \log \sum_{j=1}^{N_P} p(\hat{\mathbf{Z}}_{i:}|\hat{\mathbf{P}}_{j:})p(j)p(\hat{\mathbf{P}}_{j:}). \tag{27}$$

Here the same MLE point approximation $\hat{\rho}_{j:} = \hat{\mathbf{P}}_{j:}$ is introduced as in Eq. (21) to avoid the intractable integral. However, it is found that including the marginal term $p(\hat{\mathbf{P}}_{j:})$ in this loss function hurts performance, so instead the following is used as the clustering loss

$$\mathcal{L}_{\text{cluster}}(\hat{\mathbf{Z}}; \theta, \mathbf{Q}) := -\frac{1}{N_Z} \sum_{i=1}^{N_Z} \log \sum_{j=1}^{N_P} p(\hat{\mathbf{Z}}_{i:}|\hat{\mathbf{P}}_{j:})p(j) \tag{28}$$

$$= -\frac{1}{N_Z} \sum_{i=1}^{N_Z} \log \sum_{j=1}^{N_P} \text{VMF}_d(\hat{\mathbf{Z}}_{i:}, \hat{\mathbf{P}}_{j:}; \tau_1). \tag{29}$$

This is due to the marginal term $p(\hat{\mathbf{P}}_{j:})$ including the class prototypes $\mathbf{C}$. Removing this term from the clustering loss ensures that the prototypes $\mathbf{C}$ are primarily learnt to discriminate classes.

**Auxilary loss: Image prototype classes.** Three additional constraints are added to the model to improve quality using an auxilary loss term $L_{\text{aux}}$. For the first constraint, we observe that through Eq. (21) there is a relationship between the class prediction of the image prototypes and patches

$$p(y_l|\hat{\mathbf{Z}}) \le p(y_l|\hat{\mathbf{P}}) \tag{30}$$

where

$$p(y_l|\hat{\mathbf{P}}) := \max_j p(\nu_l|\hat{\mathbf{P}}_{j:}) \tag{31}$$

If there are no image prototypes associated with a class, the patches will also not be associated with that class. As a result, adding an additional discriminative loss term to ensure that at least one image prototypes reflects the global image label

$$\mathcal{L}_{\text{p-discrim}}(\mathbf{Z}, y; \theta, \mathbf{Q}, \mathbf{C}) := -\left( \sum_{l=1}^{c} y_l \log p(y_l|\hat{\mathbf{P}}) + (1 - y_l) \log(1 - p(y_l|\hat{\mathbf{P}})) \right), \tag{32}$$

could result in a small performance improvement.

**Auxilary loss: Image and class prototype diversity.** To ensure that the image and class prototypes have no redundancy, a second term is added to penalise duplicate image prototype vectors and class prototype vectors. This is achieved by using a contrastive loss term (Chen et al., 2020a)

$$\mathcal{L}_{\text{contrast}}(\mathbf{Z}; \theta, \mathbf{Q}, \mathbf{C}) = - \sum_i^{N_c} \log \frac{\exp(\hat{\mathbf{C}}_{i:} \cdot \hat{\mathbf{C}}_{i:}/\tau_c)}{\sum_{j=1}^{N_c} \exp(\hat{\mathbf{C}}_{i:} \cdot \hat{\mathbf{C}}_{j:}/\tau_c)}$$
$$- \sum_i^{N_P} \frac{\exp(\hat{\mathbf{P}}_{i:} \cdot \hat{\mathbf{P}}_{i:}/\tau_c)}{\sum_{j=1}^{N_P} \exp(\hat{\mathbf{P}}_{i:} \cdot \hat{\mathbf{P}}_{j:}/\tau_c)} \tag{33}$$

with the temperature parameter $\tau_c$. We note that this constraint is not memory intensive, as the contrastive loss is defined only between the prototypes that belong to a particular image.

**Auxilary loss: Image prototype consistency.** The third constraint encourages consistency in the prototypes assigned to particular patches when the image is augmented. We use a loss term inspired by Consistent Assignment for Representation Learning (Nauta et al., 2023; Silva & Rivera, 2022) (CARL)

$$\mathcal{L}_{\text{CARL}}(\mathbf{Z}; \theta, \mathbf{Q}) = - \frac{1}{N_Z} \sum_i^{N_Z} \sum_j^{N_P} \log p(\hat{\mathbf{P}}_{j:}|\hat{\mathbf{Z}}_{i:}) p(\hat{\mathbf{P}}_{j:}^*|\hat{\mathbf{Z}}_{i:}^*) \tag{34}$$

where $\mathbf{Z}^*$ and $\mathbf{P}^*$ are the patch embeddings and image prototypes from an augmented view $\mathbf{X}^*$ of the image $\mathbf{X}$. This loss term is minimised when the softmax vectors given by Eq. (19) from different views are consistent and confident. To create the augmented views we apply the same cropping and flipping augmentations, so that the patches cover the same image region, but different color and blur augmentations.

**Final training objective.** The total auxiliary loss term is given by adding these three component terms.

$$\mathcal{L}_{\text{aux}}(\mathbf{Z}, y; \theta, \mathbf{Q}, \mathbf{C}) = \mathcal{L}_{\text{p-discrim}}(\mathbf{Z}, y; \theta, \mathbf{Q}, \mathbf{C}) + \mathcal{L}_{\text{contrast}}(\mathbf{Z}; \theta, \mathbf{Q}, \mathbf{C}) + \mathcal{L}_{\text{CARL}}(\mathbf{Z}; \theta, \mathbf{Q}) \tag{35}$$

which fully specifies the complete training objective for a single image, given by

$$\mathcal{L}(\mathbf{Z}, y; \theta, \mathbf{Q}, \mathbf{C}) = \mathcal{L}_{\text{cluster}}(\mathbf{Z}; \theta, \mathbf{Q}) + \mathcal{L}_{\text{discrim}}(\mathbf{Z}, y; \theta, \mathbf{Q}, \mathbf{C}) + \mathcal{L}_{\text{aux}}(\mathbf{Z}, y; \theta, \mathbf{Q}, \mathbf{C}) \tag{36}$$

When fitting the loss over more than one image, we using a mini-batching approach and use the average over the batch as the final optimisation loss.

**Further details on label smoothing and background classes.** Label smoothing is utilised for the associations of the class prototypes $\mathbf{C}$ to the classes, as described by the matrix $\phi$. Formally, we can write this matrix as

$$\phi_{ly} := \begin{cases} 1 - \alpha + \frac{\alpha}{c} & \text{if } (l \mod \frac{N_C}{c}) = y \\ \frac{\alpha}{c} & \text{otherwise} \end{cases} \tag{37}$$

where $\alpha$ is the smoothing parameter and $c$ is the number of classes.

When including background classes, the single class label $y$ is extended into a multiclass image label by appending a 1 as the final element, meaning each image is assigned both its original class and the background class. The associations matrix is extended accordingly using the same label smoothing strategy described above, with additional entries incorporated to handle the expanded class prototypes $\mathbf{C}_{w/b} \in \mathbb{R}^{(N_C + N_N) \times d}$, where $N_C$ represents the original number of classes, $N_N$ is the number of additional background class prototypes, and $d$ is the feature dimension.

---

**Algorithm 1** Algorithm for training ComFe.

---

**Input:** Training set $T$, $N_E$ number of epochs, $f$ backbone model, Aug(.) augmentation strategy that creates two augmentations per image with the same cropping and flipping operations

Randomly initialise transformer decoder head $g_\theta$, input queries $\mathbf{Q}$, class prototypes $\mathbf{C}$ and generate class assignment matrix $\boldsymbol{\phi}$;

$i = 0$;

**while** $i < N_E$ **do**

    Randomly split $T$ into $B$ mini-batches;

    **for** $(x_b, y_b) \in \{T_1, ..., T_b, ..., T_B\}$ **do**

        $\mathbf{X} = \text{Aug}(x_b)$

        $\nu = \text{OneHot}(y_b)$

        **if** using background class prototypes **then**

            $\nu = \{\nu, [1, ..., 1, ..., 1]\}$;                        $\triangleright$ Add the background class to all images.

        **end if**

        $\mathbf{Z} = f(\mathbf{X})$;

        $\mathbf{P} = g_\theta(\mathbf{Z}, \mathbf{Q})$;

        $p(\nu|\hat{\mathbf{P}}) = \sigma\left(\hat{\mathbf{P}} \cdot \hat{\mathbf{C}}/\tau_2\right) \boldsymbol{\phi}_{:\nu}$;

        $p(\hat{\mathbf{P}}|\hat{\mathbf{Z}}) = \sigma\left(\hat{\mathbf{Z}} \cdot \hat{\mathbf{P}}/\tau_1\right)$;

        $p(\nu|\hat{\mathbf{Z}}) = \sum_{j=1}^{N_P} p(\nu|\hat{\mathbf{P}}_{j:}) p(\hat{\mathbf{P}}_{j:}|\hat{\mathbf{Z}})$;

        $p(y|\hat{\mathbf{P}}) = \text{MaxPool}(p(\nu|\hat{\mathbf{P}}))$;

        $p(y|\mathbf{Z}) = \text{MaxPool}(p(\nu|\hat{\mathbf{Z}}))$;

        Compute aux loss $\mathcal{L}_{\text{aux}} = \mathcal{L}_{\text{p-discrim}} + \mathcal{L}_{\text{contrast}} + \mathcal{L}_{\text{CARL}}$;

        Compute final loss $\mathcal{L} = \mathcal{L}_{\text{cluster}} + \mathcal{L}_{\text{discrim}} + \mathcal{L}_{\text{aux}}$;

        Minimise loss $\mathcal{L}$ by updating $\theta$, $\mathbf{C}$ and $\mathbf{Q}$;

    **end for**

    $i = i + 1$;

**end while**

---

# B  Visualising class prototypes with exemplars

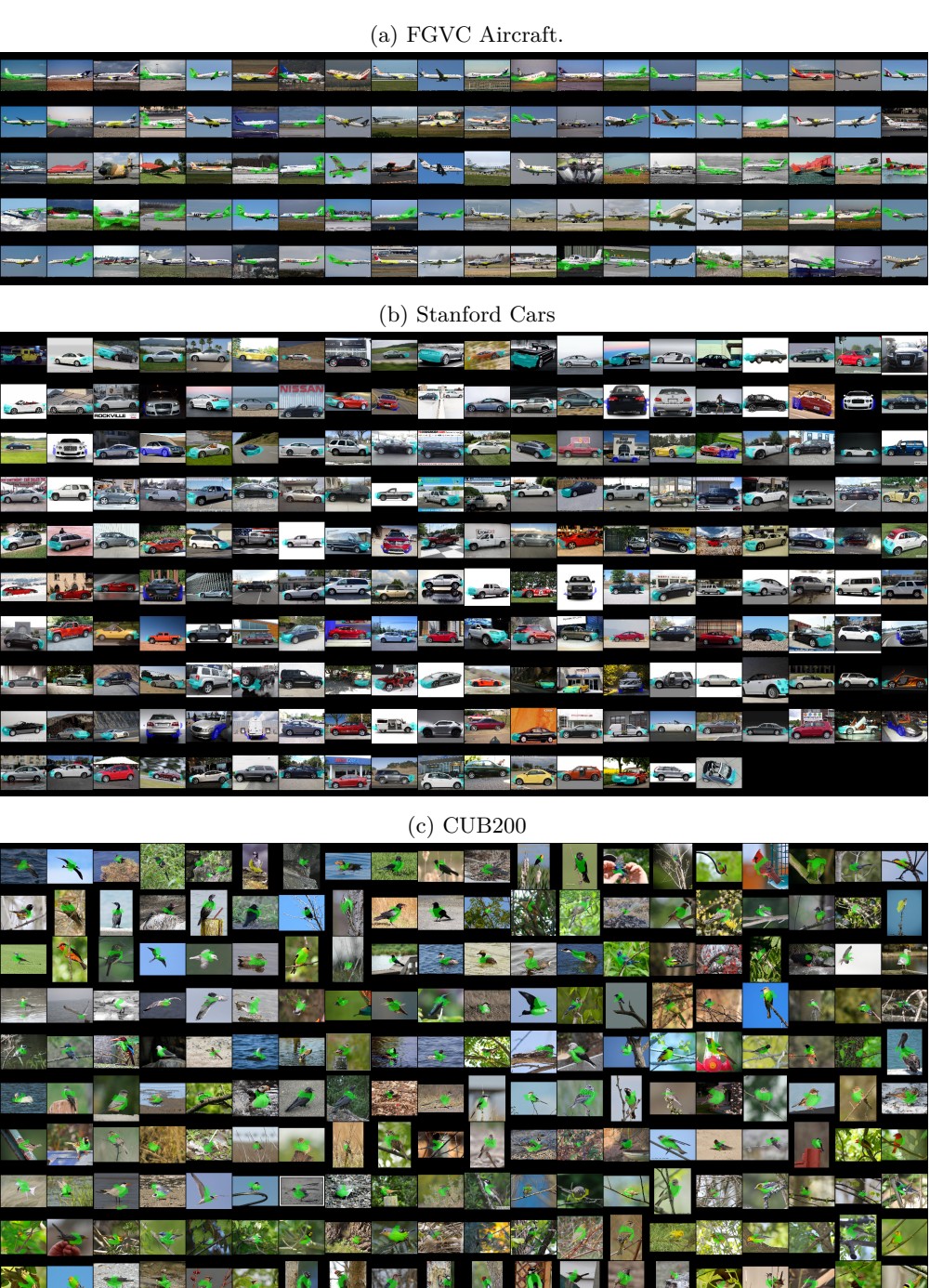

(a) FGVC Aircraft.

(b) Stanford Cars

(c) CUB200

Figure S5: **Class prototype exemplars.** Image prototypes from the training data with smallest cosine distance to the class prototypes associated with each label in the FGCV Aircraft, Stanford Cars and CUB200 datasets.

**Visualisation of class prototypes.**   In Fig. S5 we show the class prototype exemplars for all classes within the FGCV Aircraft, Stanford Cars and CUB200 datasets. The class prototype exemplars are given by the image prototypes from the training dataset with the smallest cosine distance to the class prototypes for each

label. When visualising the class prototypes in this way, we observed that not all of them are used by ComFe to make a prediction. For example, for the CUB200 dataset only one class prototype is used to learn most bird species, and the other class prototypes are located in regions with a low cosine similarity to the rest of the training dataset. For the CIFAR-10 dataset, which has a much larger number of images per class, we find that two class prototypes per class are commonly used. When the number of class prototypes are reduced in our ablation study in Section D, this has minimal impacts on the accuracy of ComFe.

(a) Ovenbird

(b) Tennessee warbler

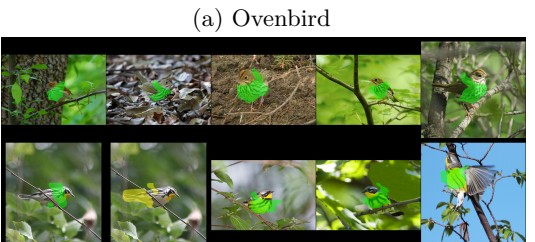
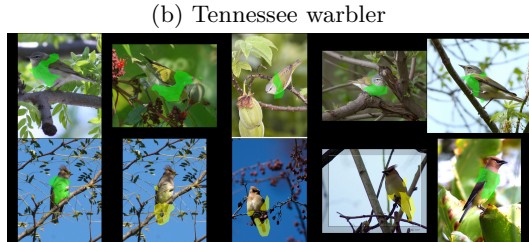

Figure S6: **Example visualizations of class prototypes and exemplars.** Five exemplars are shown for each class prototype.

**Evaluating model reasoning.**    Visualising class prototypes and exemplars illuminates the model's reasoning process. As an example, Fig. S6 shows the *Tennessee warbler* and *Ovenbird* class prototypes from the CUB-200 dataset using five exemplars. Although three class prototypes are fitted, two were relevant for classifying images for these classes, unlike most others which only required one. For the *Tennessee warbler* class, the first class prototype clearly relates to the chest and back plumage of a *Tennessee warbler*, while the second class prototype is a *Cedar waxwing*. This shows the model is likely to misclassify the latter bird, and this stems from the CUB-200 training dataset containing an image of a *Cedar waxwing* misclassified as a *Tennessee warbler*. A similar phenomenon is observed for the *Ovenbird* class, which has a spurious image in the training data of a yellow-throated warbler. Cases where spurious background features are used to classify images (Fig. S11 and Fig. S12) are also observable in this way.

## C   Additional results

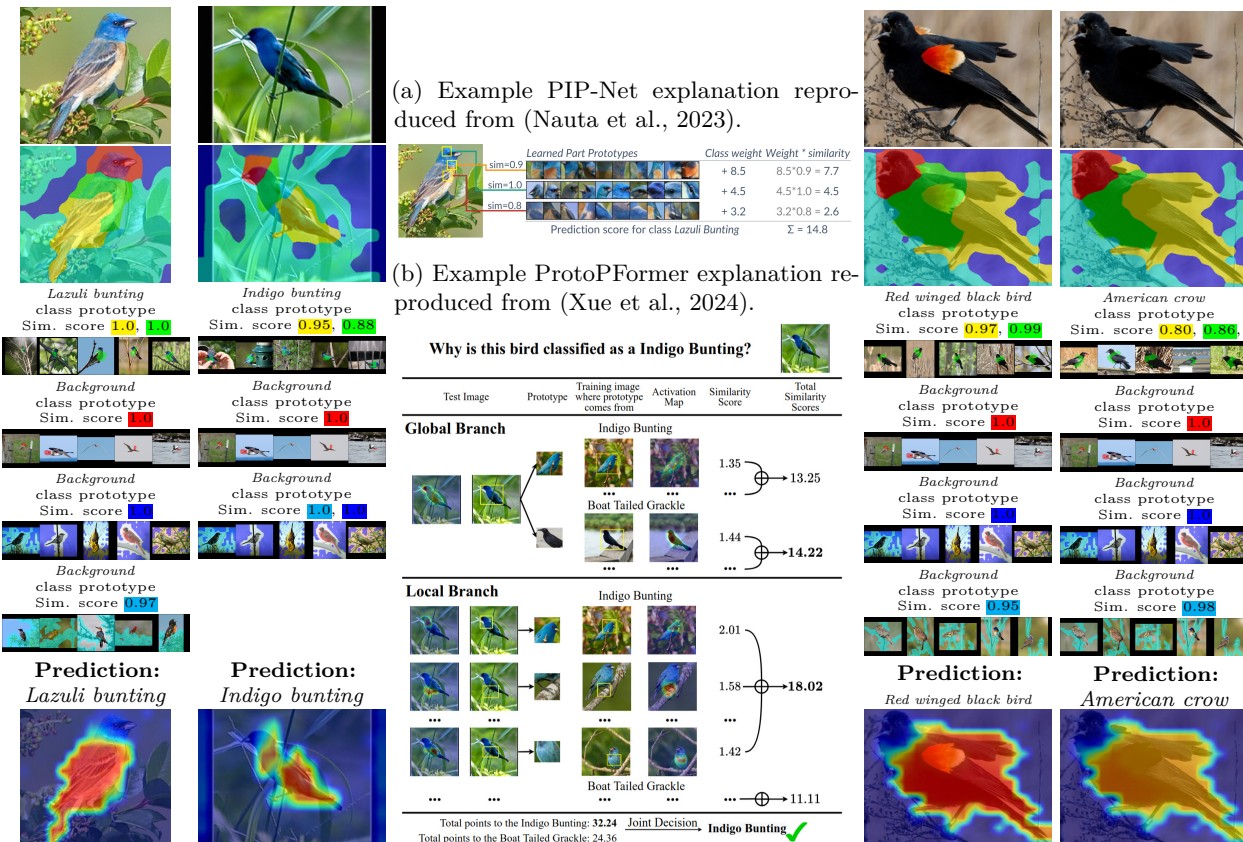

Figure S7: **Comparison of ComFe explanations versus other work.** Example of ComFe classifying a Lazuli bunting and Indigo bunting, alongside examples provided for PIP-Net and ProtoPFormer. It is also shown how modifying a red winged black bird changes the predicted class, as expected (Paul et al., 2024).

**Visual comparisons to previous interpretable frameworks.**   Fig. S7 compares ComFe explanations to that produced by PIP-Net (Nauta et al., 2023), a key inspiration for the approach. The similarity score that matches prototypes in PIP-Net is analogous to the similarity score between image prototypes and class prototypes in ComFe. While PIP-Net learns the association between prototypes and classes through the class weight matrix, ComFe assigns class prototypes to particular classes to simplify the fitting process. ComFe further provides detailed segmentation masks for the image prototypes, class prototype exemplars and prediction outputs, whereas PIP-Net is designed identify informative patches.

We also include an example for ProtoPFormer (Xue et al., 2024) in Fig. S7. The explanation provided by ProtoPFormer uses global and local branches, which are all summed together to provide an overall prediction. With the hyperparameters selected for ComFe, the scale of the ComFe explanations most similarly matches the global branch. However, finer grained explanations may be possible with ComFe by increasing the number of image prototypes or training a finer-grained backbone.

In Fig. S7 we further include an example considered in INTR (Paul et al., 2024), where an image of a *Red winged black bird* is altered to remove the red patch on the wings. In this case ComFe behaves as expected, identifying the red patch as key to predicting the *Red winged black bird* class, as per the visualised exemplars, and classifies the image as an *American crow* when the red patch is removed. This was also the behaviour observed under the INTR approach (Paul et al., 2024).

**ComFe is able to localise salient image features across a range of datasets.**   Fig. S8 shows that ComFe learns to identify relevant features for the classes of interest across a range of datasets, from small low

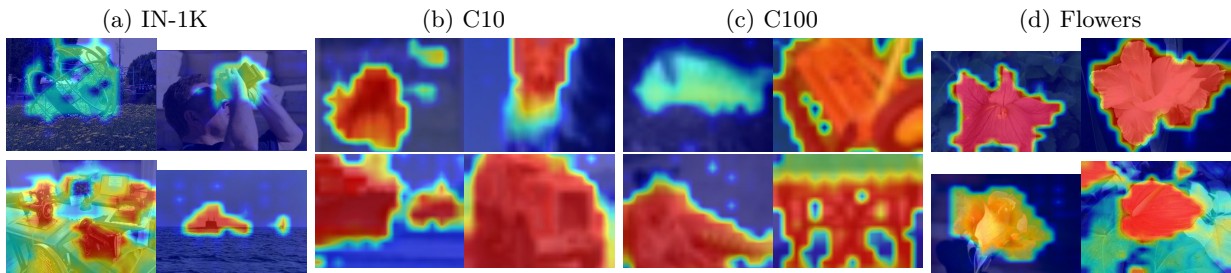

Figure S8: **Class confidence heatmaps.** Informative regions for ComFe class predictions across randomly selected validation images from ImageNet, CIFAR-10/100 and Flowers-102.

resolution datasets with few classes like CIFAR-10, to large high resolution datasets with a thousand classes such as ImageNet.

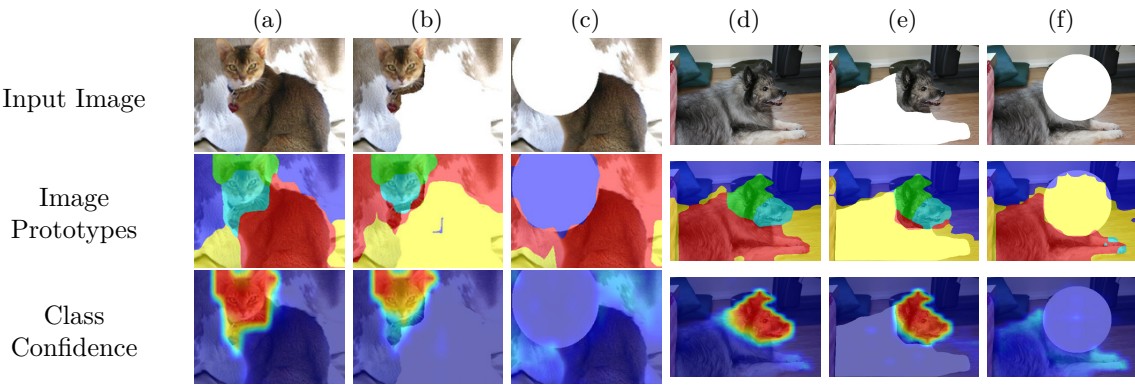

Figure S9: **Faithfulness.** Example ComFe predictions using a DINOv2 ViT-S/14 backbone from the Oxford Pets dataset that have been altered to remove the informative regions, or exclude other regions that potentially might have been informative.

**The predictions of salient image features provided by ComFe are faithful under image manipulation.** Fig. S9 shows that when parts of an image are removed that are not identified as important to classification, ComFe provides similar output. It also shows than when the regions of an image that are used to make a prediction are removed, ComFe has a low confidence that any particular class is present and will classify the image as the background class (i.e. predict that no class is present). This reflects the training process of the DINOv2 backbones, which uses a self-supervised approach that encourages patches to have similar embeddings even when other parts of the image were removed, and other ViT backbones may be less faithful.

Table S6: **Background class prototypes.** Performance of ComFe with and without background class prototypes versus a linear head.

| Head | Backbone | Background Prototypes | IN-1K | IN-V2 | Sketch | IN-R | IN-A |
|------|----------|----------------------|-------|-------|--------|------|------|
| Linear (Oquab et al., 2024) | DINOv2 ViT-S/14 (f) | | 81.1 | 70.9 | 41.2 | 37.5 | 18.9 |
| Comfe | DINOv2 ViT-S/14 (f) | 3000 | **83.0** | **73.4** | **45.3** | **42.1** | **26.4** |
| Comfe | DINOv2 ViT-S/14 (f) | 0 | 82.6 | 73.0 | 44.4 | 41.6 | 24.0 |

**Background class prototypes improve the generalisability and robustness of ComFe.** Table S6 shows that with and without background classes, ComFe obtains better performance, generalisation and robustness on ImageNet compared to a non-interpretable linear head. As shown in previous work (Jiang et al., 2021; Zhao et al., 2023), the performance and robustness of ViT models can be improved by considering the patch tokens as well as the class tokens. We also find that the use of background class prototypes slightly improves the performance of ComFe, in addition to providing a mechanism for identifying the salient regions of an image when making a prediction.

Table S7: **Registers.** Performance of ComFe on the DINOv2 backbones with (Darcet et al., 2024) and without (Oquab et al., 2024) registers.

| Backbone | Dataset | | | | | | | | |
|---|---|---|---|---|---|---|---|---|---|
| | IN-1K | C10 | C100 | Food | CUB | Pets | Cars | Aircr | Flowers |
| DINOv2 ViT-S/14 (f) | **83.0** | **98.3** | **89.2** | **92.1** | 87.6 | 94.6 | **91.1** | **77.5** | **99.0** |
| DINOv2 ViT-S/14 (f) w/reg | 82.9 | 98.2 | 89.0 | 91.6 | **87.8** | **94.9** | 90.5 | 76.5 | 98.8 |
| DINOv2 ViT-L/14 (f) | 86.7 | 99.4 | 93.6 | 94.6 | 89.2 | 95.9 | 93.6 | 83.9 | 99.4 |
| DINOv2 ViT-L/14 (f) w/reg | **87.2** | **99.5** | **94.6** | **95.6** | **90.0** | **96.0** | **94.5** | **85.6** | **99.6** |

**Including registers in the DINOv2 ViT model can improve the performance of ComFe for large models.** Further work on the ViT backbone has found that including register tokens (Darcet et al., 2024) can prevent embedding artefacts such as patch tokens with large norms, which improves results for bigger models like the ViT-L and larger variants. In Table S7 it is shown that the performance of ComFe is improved for the ViT-L model with registers. Fig. S10 shows that for the ViT-L backbone without registers, the informative patches can be irregular and not contain the relevant object of interest in the image, and the image prototypes can be less natural. However, when registers are used the image prototypes are more descriptive and we were unable to find cases where relevant features from the image were not identified by ComFe.

**For some initialisations, ComFe transparently classifies using background features.** The features that are found to be informative by ComFe can vary depending on the random seed chosen for the algorithm. As shown in Fig. S11 for the Food-101 dataset, particular initialisation seeds result in ComFe finding background features that can be used to accurately classify the training and validation data. However, there appears to be a trend across both the Food-101 (Fig. S11) and Oxford Pets (Fig. S12) datasets that model initialisations resulting in more salient image features being identified have better performance.

We observe that the DINOv2 patch embeddings are influenced by their context. For example, in Fig. S13 we use $k$-means clustering to visualise the DINOv2 patch embeddings of two images containing a white plate. We find that the surface the plates rest on both fall in the background cluster (yellow), but the plate containing churros is placed in the red cluster while the plate containing a burger is placed in the blue cluster. When ComFe obtains good performance in cases the salient parts of an image are classified as the background, this likely reflects this context leakage from the backbone model.

Prototypes being selected from the background of an image was a challenge in the original ProtoPNet approach, where a pruning process was employed to remove most of the non-informative prototypes from the reasoning process (Chen et al., 2019), and this can still occur in more recent approaches such as PIP-Net (Nauta et al., 2023).

Table S8: **Further results for alternate backbones** Results for fitting ComFe on the CUB-200 dataset with different frozen ViT-B backbones. Training approaches include supervised on IN-21K (AugReg), self-supervised on IN-1K (DINO, MAE) and foundation models (CLIP, DINOv2 and DINOv2 w/reg).

| | CLIP (Radford et al., 2021) | DINO (Caron et al., 2021) | AugReg (Steiner et al., 2022) | MAE (He et al., 2022) | DINOv2 (Oquab et al., 2024) | DINOv2 w/reg (Darcet et al., 2024) |
|---|---|---|---|---|---|---|
| CUB-200 | 83.4 | 80.2 | 85.6 | 76.9 | 88.8 | 89.4 |

(a) Class confidence    (b) Image prototypes    (c) Class confidence **w/registers**    (d) Image prototypes **w/registers**

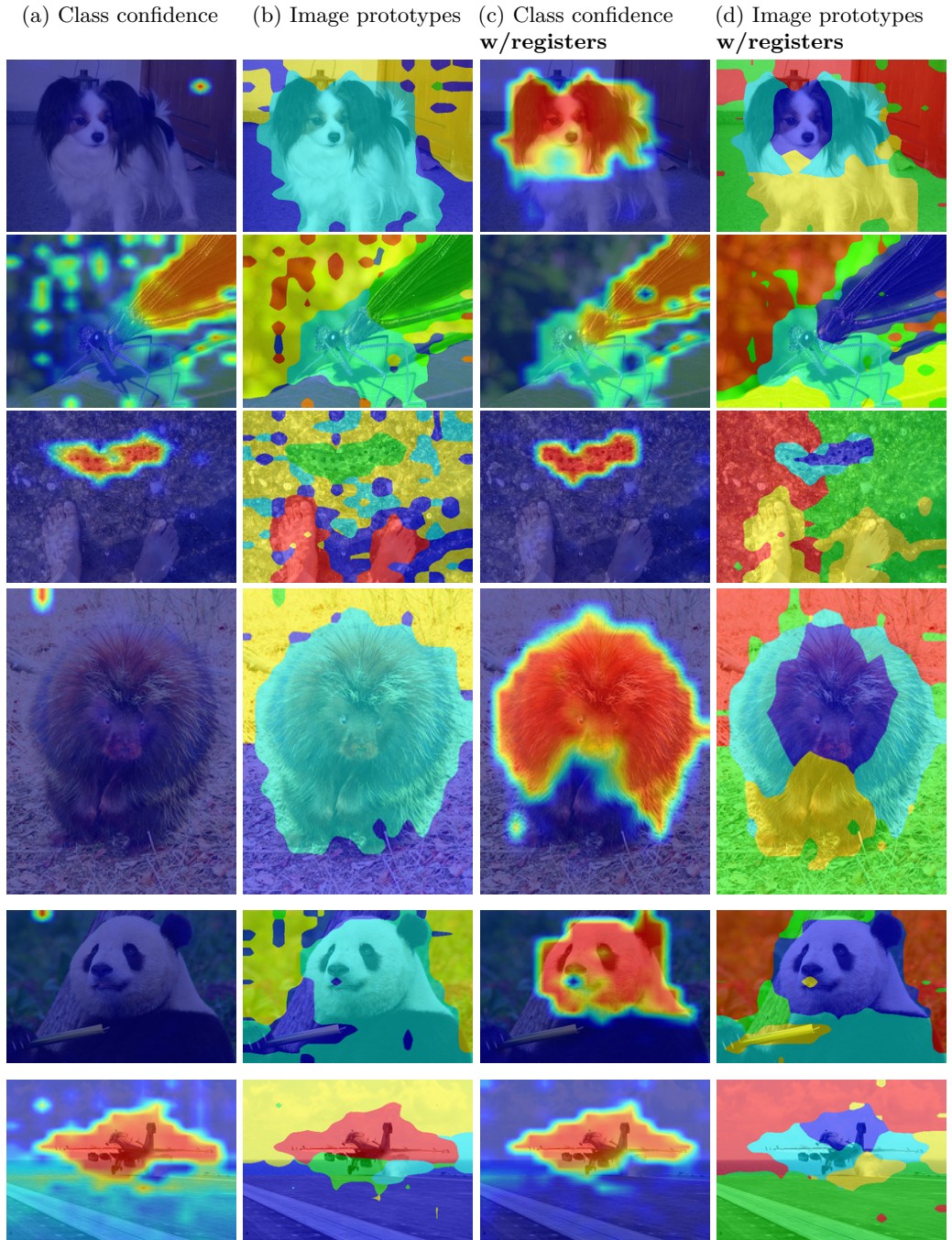

Figure S10: **Interpretability with registers.** Example ComFe image prototypes and class predictions with the DINOv2 ViT-L/14 backbone trained on ImageNet. The left two columns are obtained using the network without registers, and the right two columns use the network that includes registers.

**ComFe heads can provide interpretations for a range of different backbones.** Table S8 shows the performance of ComFe with other pretrained ViT backbones. Higher quality backbones generally perform better, and the image prototypes and features found to be informative vary between them (see Fig. S14). We note that some backbones are more likely to see background features as informative (DINO and MAE) while

| (a) Seed 1: 92.1% | (b) Seed 2: 92.2% | (c) Seed 4: 92.3% | (d) Seed 3: 92.4% |

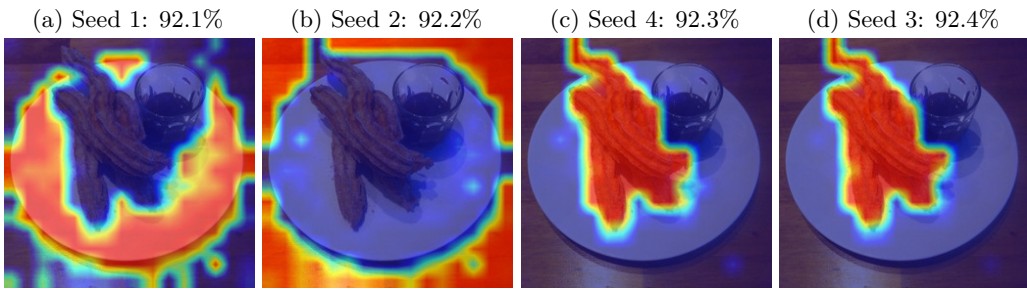

Figure S11: **Food-101 seeds.** Class confidence maps using ComFe models initialised with different seeds for an image from the Food-101 validation set.

| (a) Seed 1: 94.6% | (b) Seed 4: 94.8% | (c) Seed 3: 94.8% | (d) Seed 2: 95.3% |

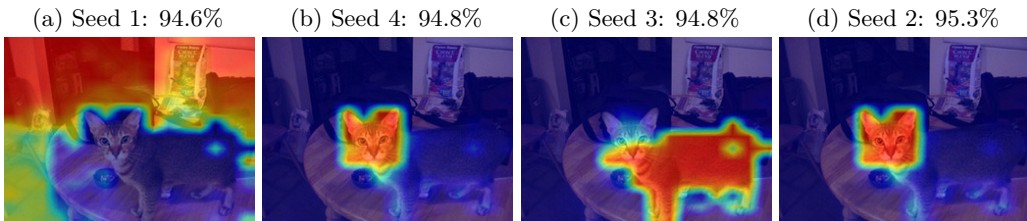

Figure S12: **Oxford Pets seeds.** Class predictions using ComFe models initialised with different seeds for an image from the Oxford Pets validation set.

| (a) Churros | (b) Clustering with DINOv2 | (c) Burger |

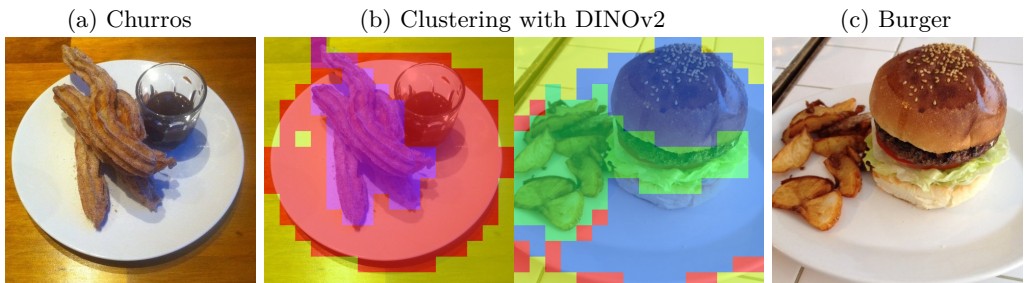

Figure S13: Example image of churros and a burger on a white plate from the Food-101 validation set. The images in the middle are overlayed with a *k*-means clustering of the combined patch embeddings of the two images.

the results for other backbones highlight artefact patches that may contain global information (Darcet et al., 2024) (AugReg and CLIP).

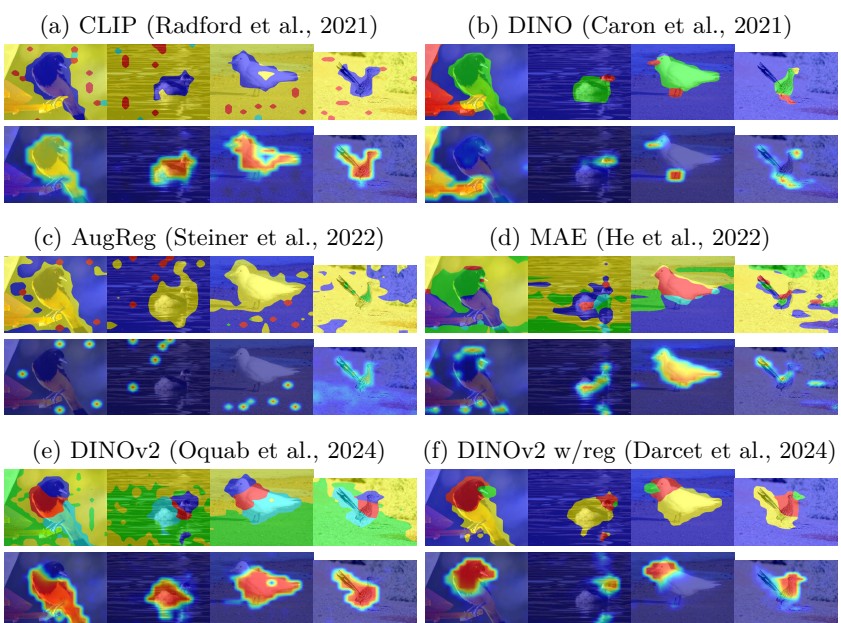

Figure S14: **Image prototypes and class confidence heatmaps.** Visualizations of the outputs of ComFe with different pretrained backbones.

# D    Ablation studies

In this section we undertake an ablation study on the hyperparameters of ComFe using the FGVC Aircraft, Oxford Pets and Stanford Cars datasets. For each set of parameters we consider five runs and report the mean and standard deviation of the validation accuracy. The bolded set of parameters reported in the tables is the default one used throughout this paper.

Table S9: **Ablation: Transformer decoder layers.** Performance of ComFe on the DINOv2 ViT-S/14 backbone with different numbers of transformer decoder layers.

| Transformer | Dataset | | |
|---|---|---|---|
| **Layers** | Aircr | Pets | Cars |
| 1 | 75.3±1.3 | 94.5±0.4 | 90.3±0.4 |
| **2** | 77.2±0.9 | 94.7±0.4 | 91.1±0.1 |
| 4 | 77.1±0.6 | 94.8±0.4 | 91.6±0.2 |
| 6 | 77.9±0.7 | 94.4±0.3 | 91.2±0.2 |

**Tranformer decoder layers.**    Table S9 shows that for the Stanford Cars and the FGVC Aircraft datasets, including more transformer decoder layers improves the average performance of ComFe across a number of seeds. However, for the Oxford Pets dataset there appears to be no significant relationship between accuracy and the size of the transformer decoder.

Table S10: **Ablation: Auxilary loss.** Performance of ComFe on the DINOv2 ViT-S/14 backbone with elements of the loss removed. In the loss term columns, an N denotes when a term is removed and a Y denotes when a term is kept.

| Loss term | | | Dataset | | |
|---|---|---|---|---|---|
| $L_{\text{p-discrim}}$ | $L_{\text{CARL}}$ | $L_{\text{contrast}}$ | Aircr | Pets | Cars |
| N | N | N | 75.5±1.3 | 94.1±0.8 | 90.3±0.5 |
| N | Y | Y | 76.4±2.2 | 94.7±0.6 | 90.5±0.7 |
| Y | N | Y | 76.8±1.0 | 94.8±0.6 | 90.9±0.5 |
| Y | Y | N | 76.7±0.6 | 94.6±0.3 | 90.8±0.3 |
| **Y** | **Y** | **Y** | 76.8±0.7 | 94.8±0.4 | 91.0±0.2 |

**Loss terms.**    In Table S10 we explore the impact of removing loss terms on the performance of ComFe across the Stanford Cars, FGVC Aircraft and Oxford Pets datasets. We find that that they have a marginal impact on the performance of the ComFe models, but do provide a small performance improvement.

Table S11: **Ablation: Temperature.** Performance of ComFe on the DINOv2 ViT-S/14 backbone with different $\tau_1$ and $\tau_2$ concentration parameters.

| Concentration | | Dataset | | |
|---|---|---|---|---|
| $\tau_1$ | $\tau_2$ | Aircr | Pets | Cars |
| 0.05 | 0.02 | 76.8±1.5 | 94.9±0.4 | 90.5±0.5 |
| 0.10 | 0.01 | 76.5±0.9 | 95.0±0.3 | 90.6±0.0 |
| **0.10** | **0.02** | 77.0±0.4 | 94.9±0.4 | 91.0±0.3 |
| 0.10 | 0.05 | 77.8±0.5 | 94.9±0.5 | 91.1±0.6 |
| 0.20 | 0.02 | 77.4±1.1 | 95.1±0.2 | 91.2±0.1 |

**Concentration parameters.** In Table S11 we explore the impact of the concentration parameters $\tau_1$ and $\tau_2$ on the performance of ComFe. While there is some scope for optimising the choice of these parameters for particular datasets, the performance of ComFe does not appear to be particularly sensitive to these parameters.

Table S12: **Ablation: Label smoothing.** Performance of ComFe on the DINOv2 ViT-S/14 backbone with different amounts of label smoothing ($\alpha$).

| Label smoothing | Dataset | | |
|---|---|---|---|
| $\alpha$ | Aircr | Pets | Cars |
| 0.0 | 77.2±0.6 | 94.8±0.3 | 91.0±0.3 |
| **0.1** | 77.3±0.7 | 94.8±0.4 | 91.0±0.2 |

**Label smoothing.** In Table S12 we explore the impact of the label smoothing parameter $\alpha$ on the performance of ComFe. We find that label smoothing has little impact on accuracy, and is not necessary for the success of the method.

Table S13: **Ablation: Number of prototypes.** Performance of ComFe on the DINOv2 ViT-S/14 backbone with different numbers of prototypes.

| Number of prototypes | | Dataset | | |
|---|---|---|---|---|
| $N_P$ | $N_C/c$ | Aircr | Pets | Cars |
| 5 | 1 | 76.7±0.7 | 94.6±0.4 | 91.0±0.4 |
| **5** | **3** | 77.6±0.8 | 94.8±0.2 | 91.0±0.3 |
| 10 | 3 | 76.1±1.2 | 94.8±0.4 | 91.3±0.3 |

**Number of prototypes.** In Table S13 we explore the impact of the number of image prototypes $N_P$ and class prototypes per label $N_C/c$ on the performance of ComFe. While there may be some marginal performance gains for particular datasets for choosing a particular set of prototypes, overall these have only a small impact on the accuracy of ComFe.

Table S14: **Subsetted robustness.** Performance of ComFe with and without background class prototypes on ImageNet robustness and generalisation benchmarks. We report accuracy with a subset of the logits, restricted to only the classes present in the ImageNet-R (Hendrycks et al., 2021a) and ImageNet-A (Hendrycks et al., 2021b) test sets.

| Backbone | Background | Test Dataset | |
|---|---|---|---|
| | Prototypes | IN-R | IN-A |
| DINOv2 ViT-S/14 (f) | 3000 | 57.7 | 42.6 |
| DINOv2 ViT-B/14 (f) | 3000 | 67.1 | 61.7 |
| DINOv2 ViT-L/14 (f) | 3000 | **72.5** | **71.9** |

