# OpenReview forum: "ComFe: An Interpretable Head for Vision Transformers"
_TMLR — Accepted by TMLR_

### Review · Reviewer_PTeC · 2025-09-08

**Summary Of Contributions:**

*Summary:*
This paper proposes ComFe an interpretable classification head for pretrained vision transformers. It represents each image by a few image prototypes and classifies the image by their similarity to learned class prototype patches. The authors show that ComFe is competitive in terms of classification accuracy with non-interpretable methods and other interpretable methods, but scales better. The authors provide a detailed ablation of their proposed method.

*Strengths:*
(S1) Interpretability is an important research direction, especially when combined with large, pretrained vision transformers. ComFe addresses this problem.
(S2) The empirical results seem convincing.
(S3) Through the reduction of the number of image patches to a fixed number of image prototypes, ComFe is memory-lightweight and scales to larger datasets than competitors.


*Weaknesses*:
(W1) No uncertainties are reported anywhere. Was the method (and the competitors) trained for multiple seeds?
(W2) The method presentation is unclear in several places (see below).
(W3) The conceptual comparison with competitors is fragmented over three places (beginning of discussion, end of section 3, section 2).
(W4) Some qualitative findings should be discussed / explored in more depth (see below).

**Additional Comments:**

C1: The caption of Tab S14 mentions ComFe without background class prototypes, but the table seems to only show ComFe with background class prototypes.

**Audience:**

Yes

**Audience Explanation:**

Making black-box neural network decision interpretable is an important research direction and crucial for trust- or safety-critical applications (S1).  Moreover, ComFe can leverage pretrained, frozen vision transformer backbones, which have recently gained a lot of attention (S1). Scaling to ImageNet-size datasets is also important (S2). Together, I am confident that the ideas of this paper will be interesting to several people in TMLR's audience.

**Broader Impact Concerns:**

I do not see any ethical implementations of the work that would require adding a Broader Impact Statement.

**Claims And Evidence:**

No

**Claims Explanation:**

While I am not an expert in the field of interpretable classification, to me the main benchmarks look convincing (both in terms of datasets and competitors) and the authors provide several ablations (S2).

However, several types of evidence are missing / should be expanded:
(E1, see W1) Uncertainties should be reported at least for ComFe in the main benchmarks to assess how large the improvements are over the variability of the method.
(E2, see W4) What can background features be used to accurately predict the classes (Fig S10, S11)? How often did this happen? The authors conjecture that a multi-head strategy might solve the issue, but do not evaluate such a strategy.
(E3, see W4) Why does the ViT-B/14 model without registers produce background artifacts, but the ViT-S/14 model without registers does not? Do the ComFe models in the main benchmarks (Tab 1, 2) perhaps also produce background artifacts.
(E4, see W4) Some explanations of ComFe seem counterintuitive and warrant some more explanation. For instance, I would have assumed that a bird's head is quite informative for its classification, but ComFe assigns the head to the background label. Could the be the case because each class only has three class prototypes and bird heads simply are not among the three most discriminative class features? A related question is why the authors chose to assign as many class prototypes to real classes as to the background. Is it because backgrounds can be varied?



(E5, see W2) I had some serious difficulty understanding the method exposition. I am sorry if this is due to a misunderstanding on my side. Equation (5) is impossible to derive without the appendix, because some key assumptions are not mentioned in the main paper, namely that the class and image prototype priors are uniform. Some of the datasets are class balanced, like CIFAR or ImageNet. Others have a strong class imbalance, like Flowers. Why can we assume uniform class prior even in these cases?

(E6, see W2) I am confused by the treatment of

 $\hat{P}_{j}$

(I'll omit the hat now as it does not get parsed properly inline) . In Eq 4 it seems as if $P_j$ is a continuous random variable supported on the hypersphere (its distribution is a mixture of von Mises-Fisher distributions). But in Eq (5) when marginalizing over $P_j$ it seems as if one only needs to marginalize over the discrete set {$1, \ldots, N_{P}$}. Which interpretation is correct? If  $P_j$  is continuous, then why do we not need to integrate over the whole hypersphere in Eq (5)? Moreover, in this case one could compute the product of two vMF distributions in closed form.

(E7, see W2) Why can we assume a uniform prior over the image prototypes $P_j$? Especially if they are continuous, their distribution is given by the class prior $p(\nu)$ and Eq (4) and we are not free to simply assume its distribution. Or does this mean that we assume a uniform prior over the set {$1, ..., N_P$}? Similarly, how can we assume the distribution of $P_j$ by Eq (4) but then compute $P_j$ via Eq (7), which most likely will not amount to a mixture of vMFs.
(E8, see W2) Even if we assume uniform class priors, I do not understand the derivation from Eq (16) to Eq (17). Each term $p(P_{j:} | \nu_l)$ depends on $\phi_{:,\nu_l}$ in Eq (16), but Eq (17) only depends on $\phi_\nu$, not on $\phi_{\nu_l}$.
(E9, see W2) I did not get Eq (34) and (35) at all.
(E10, see W2) How are the class confidence heat maps computed?
(E11, see W2) How can one compute the component feature maps like in Fig 1, 2, 3? In particular, where does the spatial information come from? If I understand Eq (7) correctly, the i-th component feature map depends on all patch embeddings of the image $Z$ jointly and not in a spatially localized way.
(E12, see W4) Even if the component feature maps (and the class confidence maps) can be computed on a patch level, why do they seem to be on pixel-level in Figs 1, 2, 3? How were the pixel-level maps computed?
(E13, see W2) How are the class prototype exemplars computed? If I understand correctly, the class prototypes have the same dimensionality as an image patch embedding. But a class exemplar is a whole image, so it has many embedded patches. Are those images selected for which one patch embedding has high cosine similarity with a class prototype? Or one for which one prototype image has high cosine similarity with the class prototype?

(E14, see W2): It would be nice to add the auxiliary losses to the main text and to perform an ablation that removes more than one of them at the same time.
(E15, see W2): I get that on datasets such as CUBS or Flowers, where each image looks structurally similar, the image prototypes can identify semantic regions such as head, wings, body of a bird and background. But on CIFAR or ImageNet the images are much more varied, yet the five queries are still shared across all images, right? Are the feature components still interpretable?


I am happy to change my evaluation in this category if (E5-E13) get resolved.

**Requested Changes:**

Crucial:
- My main concern is with the method description (E5-E13) and I think more detailed explanations / answers to my questions are critical before I can recommend acceptance.
- Uncertainties should also be reported (E1).

Nice-to-have:
- Answers to (E2-E4, E15).
- Ablations as in (E14).
- Restructuring the conceptual comparisons in on place (W3).

---

> ### Author Response · Authors · 2025-09-29
> **Response to Review**
>
> Thank you for your review of our paper. Please see our answers to your concerns below, and let us know if there is anything we can clarify further.
>
> > E1: Uncertainties should be reported for ComFe in the main benchmarks.
>
> Given that ComFe is cost-effective to train (Table 4), and that higher-performing models also yield more semantically intuitive explanations (Figures S11 and S12), a valid approach would be to train multiple ComFe heads and report the best-performing model. We avoid this strategy, as it is unnecessary to establish the advantages of ComFe. Instead, we report results from a single seed, which we find sufficient to demonstrate consistent performance gains. For completeness, we also report variability across five initializations for select datasets in the ablation study (Appendix D). These results confirm that ComFe achieves strong performance even when accounting for variation due to random initialization.
>
> > E2: Background features be used to accurately predict the classes.
>
> Instances where background features were able to predict classes (Figures S11 and S12) occurred more often with specific datasets and with particular backbones. We believe this is due to two factors: (i) the well-posedness of the classification problem, since in some datasets background cues genuinely carry discriminative information (e.g., huskies commonly photographed in snow), and (ii) backbone quality, with lower quality backbones more likely to have background features highlighted in confidence maps (see Figure S10 and S14).
>
> To clarify, we do not propose a multi-head strategy. Rather, Figures S11 and S12 illustrate a simple model-selection strategy, where choosing the best-performing model across multiple training seeds also yields semantically consistent explanations. While applying such a strategy would improve benchmark results slightly, we did not evaluate it, as the reported results which used only a single seed were already strong. We have expanded on this point in the text to make this clearer.
>
> > E3: Why does the ViT-B/14 model without registers produce background artifacts, but the ViT-S/14 model without registers does not?
>
> Prior work [1] has shown that smaller ViT models (e.g., ViT-S/14) do not require registers to avoid artifact tokens, whereas larger models (e.g., ViT-B/14) are more susceptible to learning artifact when registers are absent. We have clarified this in the text. Regarding the main benchmarks (Tables 1 and 2), ComFe models using larger ViT backbones may indeed produce background artifacts. As shown in the appendix (Table S7, Figure S10), incorporating registers in these larger models mitigates this issue and further improves results.
> [1] Darcet, Timothée, et al. "Vision transformers need registers." arXiv preprint arXiv:2309.16588 (2023).
>
> > E4: I would have assumed that a bird's head is quite informative for its classification. Why assign as many class prototypes to real classes as to the background?
>
> We thank the reviewer for this insightful point. While the ComFe model using a DINOv2 backbone visualized in Figure 3 is able to correctly classify birds even when the head is assigned to the background label, this simply reflects the embedding space of the chosen backbone model. In Figure S14, it is shown that alternative feature spaces (e.g. DINOv2 w/reg) can yield models where the head and neck of the bird is informative.
>
> Regarding prototype allocation, yes, we chose to assign an equal number of prototypes to real classes and to background, based on the expectation that background features are more varied.
>
> > E5: Equation (5) is impossible to derive without the appendix. Some of the datasets are class balanced, like CIFAR or ImageNet. Others have a strong class imbalance, like Flowers. Why can we assume uniform class prior even in these cases?
>
> We thank the reviewer for this helpful comment and have clarified this point in the text. In our formulation, we assumed uniform priors for simplicity. Extending the prior to account for class imbalance would introduce a weighting term into Equation 4, and could potentially improve performance on datasets with strong imbalance. However, this would also add complexity. We therefore adopted the uniform prior assumption throughout, which we find still provides good performance in cases with imbalanced data.
>
> Due to character limits, we will answer the next set of concerns in a new comment.

---

> ### Author Response · Authors · 2025-09-29
> **Response to Review Continued**
>
> > E6: treatment of image prototypes $\mathbf{P}$ in Eq. 4 versus Eq. 5.
>
> Within the hierarchical vMF model (Eq. 1), classes generate image prototypes (Eq. 4), and image prototypes generate patch embeddings (Eq. 3) using vMF distributions. In Eq. 5, we are tracing these relationships in reverse: given a patch embedding, we want to determine the probability that it was generated by a particular class. This is done using Bayes’ rule, and since patch embeddings and classes are not directly connected, we condition on the intermediate image prototypes.
>
> Concretely, in the line before Eq. 5, given a patch embedding, we first compute the probability that it was generated by each image prototype, and then, for each prototype, the probability that it was generated by a particular class. Both of these are categorical distributions, derived via Bayes’ rule in Appendix A. Thus, the marginalization in Eq. 5 is taken over the discrete set of prototypes rather than integrating over the continuous hypersphere.
>
> > E7: Why can we assume a uniform prior over the image prototypes? Similarly, how can we assume the distribution of $\mathbf{P}$ by Eq (4) but then compute via Eq (7).
>
> The uniform priors in Appendix A are categorical distributions, and we have added a note to emphasize this and improve clarity. Eq. 4 describes the conditional distribution of the image prototypes given a class. In contrast, the prior distributions in Appendix A describe the probability that a particular image prototype generates a patch before observing any data. We assume a uniform prior in this context as a neutral zero-case guess: initially, any prototype is equally likely to generate a patch.
>
> Eq. 7 (now 8) describes how the image prototypes are generated for each image using a transformer decoder, where the parameters are learned by minimizing the loss function, which incorporates the distribution in Eq. 4 as a key part in the derivation of the objective.
>
> > E8: Even if we assume uniform class priors, I do not understand the derivation from Eq (16) to Eq (17).
>
> Eq. 17 (now 18) comes from inserting Eq. 4 into Eq. 16 (now 17), and simplifying. This is where the $\phi$ term comes from. We have noted this in the text to make it clearer.
>
> > E9: I did not get Eq (34) and (35) at all.
>
> These described how the background classes were incorporated. We have redrafted this section to make it clearer.
>
> > E10: How are the class confidence heat maps computed?
>
> The class confidence heatmaps are calculated using Eq. 5 for the class that has highest confidence. We added this to the new Figure 3 to make this clearer.
>
> > E11: How can one compute the component feature maps? Where does the spatial information come from?
>
> The component feature maps are computed using Eq 15, and are colored according to the image prototype that has greatest probability for each patch. The spatial information comes from each patch having a likelihood of being generated from each image prototype. We have also added this to Figure 3 to make this clearer.
>
> > E12: Even if the component feature maps (and the class confidence maps) can be computed on a patch level, why do they seem to be on pixel-level?
>
> The patch-level (16x16) results are upsampled onto the pixel level (224x224) using bilinear interpolation. We have added this detail to the paper at the end of Section 4.1.
>
> > E13: How are the class prototype exemplars computed?
>
> The class prototype exemplars for a particular class prototype are the image prototypes within the training dataset that have highest cosine similarity to that class prototype, as calculated using Eq. 12. These image prototypes are shown by the colored masks on the class prototype exemplar images. We have edited the section on page 6 that explains the class prototype exemplars to make this clearer.
>
> > E14: Perform an ablation that removes more than one auxiliary loss at the same time.
>
> We have kept them separate from the main text, as they are not necessary for the model. We have added an ablation which removes all auxiliary losses in Table S10, which shows they marginally improve performance but are not necessary.
>
> > E15: five image prototypes are shared across all images, right? Are the feature components still interpretable for CIFAR/ImageNet?
>
> Yes, the five queries are shared across all images. We visualise the image prototypes for ImageNet-1K in Figure S10, and we find that they can still be consistently interpretable using a DINOv2 ViT-L/14 backbone with registers.

---

> > ### Comment · Reviewer_PTeC · 2025-10-09
> >
> > I thank the authors for the detailed and helpful reply! Most points were satisfyingly addressed. The following issues were not resolved yet:
> >
> > > E1: Uncertainties
> >
> > Thanks for pointing me to the uncertainty results in appendix D. I'd recommend to include them and uncertainties for the other datasets for ComFe in Tables 1 and 2. In general, I strongly think that running a new proposed method for multiple seeds and reporting means and uncertainty is very important. Running for multiple seeds and picking the best performing method feels like cherry picking and should be avoided in my mind.
> >
> > > E6: Treatment of $\hat{P_{j:}}$
> >
> > My point was that if the generative model is $\nu \to P_j \to Z_i$, then applying Bayes rule should yield
> >
> > $$P(\nu | Z_i) = \sum_j \int_{P_j\in \mathbb{S}^d} P(\nu | P_j) P(P_j | Z_i)$$
> >
> > So, why is there no marginalization over $P_j \in \mathbb{S}^d$?
> >
> > >E7 Priors over image prototypes
> >
> > Thanks, this addresses my concern partly. But I still do not see how you can on the one hand place a mixture of vMF distribution on $P_j$ in Eq (4), but on the other hand compute $P_i$ in a specific way in Eq (7). Will Eq (7) match the distribution $P(P_i | Z_i)$ that Equations 3, 4, and the Bayes rule impose for $P_i$?
> >
> > >E8 Eq 16/17
> >
> > Thanks for the comment. After thinking about it some more, I think my missing link was that $\sum_{v=1}^c \phi_{i, v} = 1 $ for all $i$.

---

> ### Author Response · Authors · 2025-10-11
> **Replying to Response**
>
> Thank you for raising these remaining issues. Please see our response below, and let us know if we need to expand further.
>
> > E1: Uncertainties
>
> For clustering algorithms that are computationally inexpensive to fit but sensitive to the initial starting point, it is common practice to run the algorithm multiple times with different initializations and select the clustering that maximizes a particular metric of clustering quality [1].
>
> [1] Jain, Anil K. "Data clustering: 50 years beyond K-means." Pattern recognition letters 31.8 (2010): 651-666.
>
> The same principle applies to ComFe, which can also be interpreted as a clustering method. In the time it takes to fit a single PIP-Net or ProtoPFormer model, multiple ComFe models can be trained (Table 4). Given ComFe's efficiency, in practical settings one should fit multiple ComFe models and select the one that performs best (with the best explanation), as illustrated in Figures S11 and S12.
>
> To maintain consistency between the reported performance in Tables 1 and 2 and the example explanations in Figure 4 and elsewhere, we report results for a single (non-cherry picked) seed, while noting that variability due to initialization is further explored in the Supporting Information.
>
> > E6: Treatment of $P_j$
>
> Thank you for raising this point. You are correct. We have updated Eq. 19 and clarified that we avoid the challenging integral by approximating it using the directions $P_j$ that maximize the model likelihood, as generated through Eq. 8.
>
> > E7: Justification for Equation 7 (now 8).
>
> Yes, the idea is that once the model is trained, Eq. 8 will match these distributions. The calculation of $P_j$ in Eq. 8 is an example of amortized inference [2]. The distribution of $P_j$ depends on both the classes and the specific image. In a traditional approach, this would require learning a separate set of image-specific prototypes $P_j$ for each image. Instead, we adopt an amortized strategy, where the parameters $P_j$ are predicted by the function in Eq. 8 (parameterized by $\theta$ and optimized using a loss derived from Eqs. 3 and 4). This approach makes the method both computationally tractable and generalizable to images outside the training dataset. We have added this detail to the paper.
>
> [2] Margossian, Charles C., and David M. Blei. "Amortized variational inference: when and why?." Proceedings of the Fortieth Conference on Uncertainty in Artificial Intelligence. 2024.
>
> > E8: Eq 16/17
>
> Yes, that is correct.

---

> > ### Comment · Reviewer_PTeC · 2025-10-16
> >
> > Thanks for these additional clarifications!
> >
> > > E1: Uncertainties
> >
> > I agree that it can make sense to rerun cheap clustering methods several times and choosing the run with lowest loss, as is commonly done for k-means. Nevertheless, the spread of individual runs is an important piece of information when first introducing such a procedure and I would have preferred it being featured in the main paper. Nevertheless, the current discussion in the limitations section is acceptable.
> >
> >
> > > E7: Justification for Equation 7 (now 8).
> >
> > Thanks for clarifying that amortised inference is used. To clarify further, the amortization is for $\nu$ only (in whose distribution $P_j$ serve as parameters), not for $P_j$ itself (which is also a latent variable, not just a parameter of $\nu$'s distribution), right? In particular, Eq (8) is a deterministic map from $Z$ to $P_j$. So it cannot ever, even after training, match the non-delta distribution $p(P_j | Z)$ that Eq (3,4) and the Bayes rule imply. The former is deterministic, the latter is not.
> >
> > Moreover, there are still other aspects of the presentation of the probabilistic framework that need improvement, I think. For instance, the authors added a statement
> >
> > > We assume uniform categorical distributions for the priors $p(P_j)$  ...
> >
> > But as mentioned above $p(P_j)$ has continuous support on $\mathbb{S}^d$. Perhaps the authors mean the uniform categorical distribution over $1, ..., N_P$? Similarly at the beginning of Appendix A.
> >
> > Conversely, after Eq (26) the authors write
> >
> > > with the constant C resulting from the uniform prior on the image prototypes $P_j$
> >
> > But here $p(P_j)$ is a distribution over the sphere $\mathbb{S}^d$, not the as uniform assumed categorical distribution over $1, ..., N_P$. In particular, $p(P_j)$ is given by Eq (4) and the uniformity of $p(\nu)$ and itself not uniform, but a mixture of VMF distributions. This also shows that the clustering loss depends on $C$, because $p(P_j|\nu)$ does.
> >
> > Without a clear description of the probabilistic setup, its the assumptions, and implementation, I have trouble judging the correctness of this work.

---

> > > ### Author Response · Authors · 2025-10-17
> > > **Response to Review**
> > >
> > > Thank you for raising these issues. We've amended the manuscript as noted below.
> > >
> > > > E7: Justification for Equation 7 (now 8).
> > >
> > > As noted below Eq. 20, we use a maximum likelihood point estimate for $P_j$ throughout the paper, rather than modeling its full distribution. While this introduces an approximation (as noted below Eq. 20), it is necessary to maintain the efficiency of ComFe. We have amended the text to clarify this, moving the section on parameterizing the prototypes to appear after the model specification and before the likelihood.
> > >
> > > The amortization in our approach refers specifically to $P_j$ itself. To avoid potential confusion, we have updated the wording to “amortization-inspired'' since we do not model the full distribution of $P_j$, but adopt a key idea of amortization: generating parameters of a distribution via a learned function using the input data.
> > >
> > > Where the manuscript previously referred to $p(P_j)$, we have replaced this with $p(j)$ to clarify that this denotes the uniform categorical distribution over $\{1,\dots,N_P\}$, and not a prior. Thank you for pointing out that $p(P_j)$ is not a prior but a marginal, as per Eq. 1. This discussion highlights that our fitting approach introduces additional approximations to the vMF mixture model to enable stable and efficient training of ComFe. Specifically:
> > >
> > > - In computing $p(P_j|Z_i)$, we approximate $p(P_j)$ with a uniform distribution over the sphere so that $p(P_j|Z_i)$ becomes straightforward to compute and independent of $\nu$.
> > > - In the clustering loss, we also employ a maximum likelihood estimate for $P_j$ to avoid the integral, and the marginal $p(P_j)$ is omitted to improve performance.
> > >
> > > We have amended the text in Appendix A to reflect these points. While these approximations are significant, they accurately reflect how ComFe is trained and the empirical results highlight the efficacy of the approach.

---

> > > > ### Comment · Reviewer_PTeC · 2025-10-17
> > > >
> > > > Thanks for clarifying your approach! I have no further questions.

---

### Review · Reviewer_aikr · 2025-09-13

**Summary Of Contributions:**

It introduces a classifier head that extracts image prototypes on the fly and compares them with learned class prototypes to build a scalable interpretable classifier. The method is then compared against other prototype-based approaches as well as non-interpretable baselines.

**Audience:**

Yes

**Audience Explanation:**

Since the classifier head can be added to any existing ViT, this work should be of interest to researchers focused on interpretability and vision transformers.

**Claims And Evidence:**

Yes

**Claims Explanation:**

The methodology, particularly the use of the vMF distribution for hierarchically modeling the prototypes/patches, feels novel. Since the core idea is scalability, the authors demonstrate that ComFe achieves comparable performance to prototype-based methods while requiring significantly less storage.

**Requested Changes:**

The authors claim that previous methods require hyperparameter tuning across datasets, whereas ComFe does not. However, doesn’t this also apply to ComFe? Both $N_P$ (the number of image prototypes) and $N_c$ (the number of class prototypes per class) appear to be dataset-dependent. Intuitively, more complex datasets would require larger values, while simpler datasets might work with fewer.

---

> ### Author Response · Authors · 2025-09-28
> **Response to Review**
>
> Thank you for your review of our paper. Please see our answers to your concerns below, and let us know if there is anything we can clarify further.
>
> > RC1: The authors claim that previous methods require hyperparameter tuning across datasets, whereas ComFe does not.
>
> Thank you for raising this point. While it is true that the number of image prototypes and class prototypes could, in principle, be dataset-dependent, in practice we found that a simple heuristic yields strong performance across diverse datasets. Specifically, we used five image prototypes and six class prototypes per class (three assigned to each class and the remaining to the background class), as described in Section 4.1. This was applied uniformly across all nine benchmark datasets in Table 1, without dataset-specific tuning. We have edited the text to make this clearer.
>
> Further ablation studies on how performance changes with different heuristics are provided in Table S13.

---

### Review · Reviewer_cAqX · 2025-09-23

**Summary Of Contributions:**

This paper proposes a novel classifier head that can be attached to frozen, pretrained vision transformers, offering both improved classification accuracy and inherent interpretability. The authors ground their method in statistical theory which expands upon prior Proto approaches and demonstrate the efficacy of their approach with multiple key vision datasets, including ImageNet-1k. Additionally, the authors explore interpretability with compelling examples and corresponding in-depth analysis.

Strengths:

-	The proposed method is well motivated and offers a reasonable approach to training dataset specific classifier heads on frozen ViT backbones.
-	The interpretability results are compelling, demonstrating applicability to multiple model types and scales.
-	Performance is rigorously evaluated, for both classification and interpretability, covering multiple benchmarks.

Weaknesses:

-	Some aspects of the paper were unclear or missing context.
-	The initialization strategy is not well described, which is significant due to the variation observed with training seeds.
-	The approach is brittle, showing sensitivity to initial conditions.

**Additional Comments:**

Questions:

-	The interpretability seed sensitivity examples demonstrate similar classification performance with less informative or nonsensical confidence maps. Does this undermine the significance of an interpretable method, or does it lead to clear sample-specific classification errors?
-	There was some discussion in the paper about training multiple heads and picking the best one to address the initialization sensitivity. Can class entries be stitched together from multiple training runs? Otherwise, could there be cases where a more variable dataset such as ImageNet results in some class confidence maps having strong semantic overlap, while others do not?
-	The DINOv2 models with registers show better prediction accuracy, but lower completeness. Is this a desirable aspect for an interpretable model, or largely a consequence of the global classification task? For example, detecting the full foreground object may make more sense to a human observer, but appears to not be necessary for classification.

**Audience:**

Yes

**Audience Explanation:**

The interpretability results are compelling, with many visual examples and in-depth analysis. The authors also discuss how the heads learn to classify class components, which do not end up contributing to the final classification; an interesting result. Furthermore, the authors describe how the interpretability analysis led to the discovery of misclassified samples in the CUB-200 dataset.

**Broader Impact Concerns:**

No concern.

**Claims And Evidence:**

Yes

**Claims Explanation:**

The authors successfully demonstrate matching or improved performance compared with prior approaches, both for classification and interpretability. Evidence is supported by top-1 accuracy, visual confidence maps, and evaluation on FunnyBirds.

**Requested Changes:**

-	The introduction should more clearly state that ComFe is primarily a classifier head exhibits interpretability, rather than a post-training interpretability technique. In some sub-fields, interpretability heads are trained to understand a models behavior without being used for downstream predictions.
-	The head architecture should be better described. The authors state that decoder layers with cross-attention are used, but seeing a more specific diagram would be helpful to understanding the activation flow, and what role Q and C play at a glance.
-	The #P column for the linear methods should be included as they are not part of the DINOv2 backbone.
-	The evaluation metric (i.e. top-1) should be clearly stated in the comparison tables – this is only apparent when considering the broader context.
-	The training protocol should be better described. How are the class exemplars and background images chosen? Are they per batch or static for the training run? Is the seed sensitivity based on the parameter initialization, or is it also susceptible to the dataset shuffling?
- It would be helpful to understand if ComFe is also applicable to semantic segmentation, especially since certain semantic prototypes did not inform the classification and are instead treated as background, leading to incomplete confidence maps. Semantic segmentation on the other hand would penalize such mapping.

---

> ### Author Response · Authors · 2025-09-28
> **Response to Review**
>
> Thank you for your review of our paper. Please see our answers to your concerns below, and let us know if there is anything we can clarify further.
>
> > AC1. Seed sensitivity can demonstrate similar classification performance with less informative confidence maps.
>
> From our experience in other contexts, seeds that result in models that rely heavily on background features may achieve strong accuracy on train/test splits but generalize far more poorly than seeds that train models whose predictions align with meaningful semantic features. This does not undermine the value of the method, but rather shows it functioning as intended. The likelihood a ComFe model produces nonsensical confidence maps reflects both the dataset quality and the backbone quality. In some datasets, background cues genuinely carry discriminative information (e.g. cats being photographed inside in particular settings), while poorer quality backbones are more likely to produce less meaningful confidence maps (Figure S10 and S13).
>
> > AC2. Stitching class entries from multiple training runs.
>
> While class entries could be stitched together from multiple training runs, this will make interpretability more complex. We simply suggest the approach taken in Figure S11 and S12, where multiple models are trained and the highest performing one (also with the best confidence map) are selected to be used. We have expanded the discussion to make this clearer.
>
> Figure S10 shows that the quality of class confidence maps relates to the quality of the pretrained ViT backbone. In this figure, for ImageNet, when the ViT-B/14 backbone is used, some confidence maps for particular classes are a single background patch. This is rectified when using the ViT-B/14 backbone with registers, which produces more semantically meaningful maps.
>
> > AC3. Backbones with registers have lower completeness.
>
> Whether this is desirable for an interpretable model depends on the context. Completeness measures the fidelity of the class confidence map—all important parts should be highlighted, preserving important parts should result in the same prediction, and removing important parts should result in a different prediction. This trades off against distractablitity—the degree the class confidence map focuses on irrelevant parts. While the ViT-S/14 backbone achieves highest completeness, highlighting the full foreground object, the ViT-B/14 backbone with registers achieves a better distractability score, highlighting only particular parts that are found to be informative for the prediction.
>
> > RC1. The introduction should more clearly state that ComFe is primarily a classifier head exhibits interpretability.
>
> Thank you for this point. We added a clarifying statement on this to the introduction.
>
> > RC2. The head architecture should be better described.
>
> We have added Figure 3 to better describe the head architecture, and make following key parts of the method easier.
>
> > RC3. The #P column for the linear methods should be included as they are not part of the DINOv2 backbone.
>
> These are negligible in comparison to the number of parameters in the head (less than 1 million). We have added this to the tables.
>
> > RC4. The evaluation metric (i.e. top-1) should be clearly stated in the comparison tables.
>
> This has been added to the tables in the main text.
>
> > RC5. How are the class exemplars and background images chosen? The training protocol should be better described.
>
> In Section 3, under the “Class Prototype Exemplars” heading, we describe how exemplars are selected. We have clarified that these exemplars are used only to interpret the class prototypes after training and are not used during the training process.
>
> We have also revised the text on page 20 regarding how the background class is included to make this process clearer to the reader.
>
> Regarding seed sensitivity, it is most likely due to the initialization of the class prototypes and clustering head, but could be impacted by dataset shuffling/batch size to some extent. We have added additional details in Section 4.1 describing the initialization strategy.
>
> > RC6. Applicability to semantic segmentation
>
> Yes, the ComFe loss could be adapted to use segmentation masks where this is available. We have added a point to the discussion covering this.

---

### Decision · Action_Editor_zQwX · 2025-10-29

**Recommendation:** Accept as is

**Audience:**

Yes

**Audience Explanation:**

The findings of this paper on its interpretable classification head for frozen vision transformer backbones, with scalability to large datasets like ImageNet-1K, is certainly of interest to a segment of the TMLR audience. The three reviewers and AE concur.

**Claims And Evidence:**

Yes

**Claims Explanation:**

The main claims regarding scalability to larger datasets, competitive performance with non-interpretable and previous interpretable methods, and improved interpretability are well supported by quantitative experimental results and visualizations. The reviewers and AE are all in agreement that the paper has been adequately validated.